# Event generation and statistical sampling for physics with deep generative models and a density information buffer

Sydney Otten [1,2 ✉], Sascha Caron[1,3], Wieske de Swart [1], Melissa van Beekveld[1,3], Luc Hendriks[1], Caspar van Leeuwen [4], Damian Podareanu[4], Roberto Ruiz de Austri[5] & Rob Verheyen[1]

Simulating nature and in particular processes in particle physics require expensive computations and sometimes would take much longer than scientists can afford. Here, we explore ways to a solution for this problem by investigating recent advances in generative modeling and present a study for the generation of events from a physical process with deep generative models. The simulation of physical processes requires not only the production of physical events, but to also ensure that these events occur with the correct frequencies. We investigate the feasibility of learning the event generation and the frequency of occurrence with several generative machine learning models to produce events like Monte Carlo generators. We study three processes: a simple two-body decay, the processes $e^+e^- \to Z \to l^+l^-$ and $pp \to t\bar{t}$ including the decay of the top quarks and a simulation of the detector response. By buffering density information of encoded Monte Carlo events given the encoder of a Variational Autoencoder we are able to construct a prior for the sampling of new events from the decoder that yields distributions that are in very good agreement with real Monte Carlo events and are generated several orders of magnitude faster. Applications of this work include generic density estimation and sampling, targeted event generation via a principal component analysis of encoded ground truth data, anomaly detection and more efficient importance sampling, e.g., for the phase space integration of matrix elements in quantum field theories.

[1] Institute for Mathematics, Astro- and Particle Physics IMAPP Radboud Universiteit, Nijmegen, The Netherlands. [2] GRAPPA, University of Amsterdam, Amsterdam, The Netherlands. [3] Nikhef, Amsterdam, The Netherlands. [4] SURFsara, Amsterdam, The Netherlands. [5] Instituto de Fisica Corpuscular, IFIC-UV/CSIC University of Valencia, Valencia, Spain. ✉email: Sydney.Otten@ru.nl

The simulation of physical and other statistical processes is typically performed in two steps: first, one samples (pseudo)random numbers; in the second step an algorithm transforms these random numbers into simulated physical events. Here, physical events are high energy particle collisions. This is known as the Monte Carlo (MC) method. Currently, a fundamental problem with these numerical simulations is their immense need for computational resources. As such, the corresponding scientific progress is restricted due to the speed of and budget for simulation. As an example, the full pipeline of the MC event generation in particle physics experiments including the detector response may take up to 10 min per event[1–7] and largely depends on non-optimal MC sampling algorithms such as VEGAS[8]. Accelerating the event generation pipeline with the help of machine learning can provide a significant speed up for signal studies allowing e.g., broader searches for signals of new physics. Another issue is the inability to exactly specify the properties of the events the simulation produces. Data analysis often requires the generation of events which are kinematically similar to events seen in the data. Current event generators typically accommodate this by generating a large number of events and then selecting the interesting ones with a low efficiency. Events that were not selected in that procedure are often discarded. It is of interest to investigate ways in which the generation of such events can be avoided.

Most of the efforts of the machine learning community regarding generative models are typically not directly aimed at learning the correct frequency of occurrence. So far, applications of generative ML approaches in particle physics focused on image generation[9–12] due to the recent successes in unsupervised machine learning with generative adversarial networks (GANs)[13–15] to generate realistic images according to human judgment[16,17]. GANs were applied to the simulation of detector responses to hadronic jets and were able to accurately model aggregated pixel intensities as well as distributions of high level variables that are used for quark/gluon discrimination and merged jets tagging[18]. The authors start from jet images and use an Image-to-Image translation technique[19] and condition the generator on the particle level content.

Soon after the initial preprint of the present article, two relevant papers appeared that model the event generation with GANs[20,21]. There the authors have achieved an approximate agreement between the true and the generated distributions. Since those papers looked at processes involving two objects such that the generator output was 7 or 8 dimensional, it is still an open question which generative models are able to reliably model processes with a larger number of objects. In addition, in both papers the authors report difficulties with learning the azimuthal density $\phi$ which we also target in our studies. In[20] the authors circumvent the trouble of learning $\phi$ explicitly with their GAN by learning only $\Delta\phi$, manually sampling $\phi_{j_1}$ from a uniform distribution and processing the data with an additional random rotation of the system. This further reduces the dimensionality of the studied problems.

In this article we outline an alternative approach to the MC simulation of physical and statistical processes with machine learning and provide a comparison between traditional methods and several deep generative models. All of these processes are characterized by some outcome $\mathbf{x}$. The data we use to train the generative models is a collection of such outcomes and we consider them as samples drawn from a probability density $p(\mathbf{x})$. The main challenge we tackle is to create a model that learns a transformation from a random variable $\mathbf{z} \rightarrow \mathbf{x}$ such that the distribution of $\mathbf{x}$ follows $p(\mathbf{x})$ and enables us to quickly generate more samples.

We investigate several GAN architectures with default hyperparameters and Variational Autoencoders (VAEs)[22] and provide more insights that pave the way towards highly efficient modeling of stochastic processes like the event generation at particle accelerators with deep generative models. We present the B-VAE, a setup of the variational autoencoder with a heavily weighted reconstruction loss and a latent code density estimation based on observations of encoded ground truth data. We also perform a first exploration of its hyperparameter space to optimize the generalization properties.

To test our setup, three different types of data with increasing dimensionality and complexity are generated. In a first step we construct generative models for a 10-dimensional two-body decay toy-model and compare several distributions in the real MC and the generated ML model data. We confirm the recent findings that both GANs and VAEs are generally able to generate events from physical processes. Subsequently, we study two more complex processes:

- the 16-dimensional $Z$ boson production from $e^+e^-$ collisions and its decay to two leptons, $e^+e^-$ and $\mu^+\mu^-$, with four 4-vectors per data point of which two are always zero.
- the 26-dimensional $t\bar{t}$ production from proton collisions, where at least one of the top quarks is required to decay leptonically with a mixture of five or six final state objects.

The study on $Z$ bosons reveals that standard variational autoencoders can't reliably model the process but confirms good agreement for the B-VAE.

For $t\bar{t}$ we find that by using the B-VAE we are able to produce a realistic collection of events that follows the distributions present in the MC event data. We search for the best B-VAE architecture and explore different possibilities of creating a practical prior by trying to learn the latent code density of encoded ground truth data. We also present results for several GAN architectures with the recommended hyperparameters.

We perform a principal component analysis (PCA)[23,24] of encoded ground truth data in the latent space of the VAE for $t\bar{t}$ production and demonstrate an option to steer the generation of events. Finally, we discuss several further applications of this work including anomaly detection and the utilization for the phase space integration of matrix elements.

In short, the structure of the paper is as follows: In "Results" we present the results. We show

- the two-body decay toy model and the leptonic Z-decay,
- the $t\bar{t} \rightarrow 4j + 1\text{or}2l$, where we optimize for several hyperparameters, assess different ways of utilizing latent code densities and show how several GAN architectures with default hyperparameters perform and
- two sanity checks on $t\bar{t}$: (a) Gaussian smearing, creating Gaussian Mixture Models and Kernel Density Estimators for events and (b) investigating whether the B-VAE learns the identity function.

"Discussion" provides the discussion, including several applications and conclusions. In "Methods" we briefly explain how we create the training data and we present the deep learning methodology. We

- provide a brief overview of GANs,
- explain VAEs and our method, the B-VAE,
- present several methods to assess the density of the latent code of a VAE and

- define figures of merit that are used to evaluate our generative models.

The main achievement of this investigation is the B-VAE and its fine-tuning whose performance is shown on several datasets and compared to other recent generative models in "Results". We find that the B-VAE is able to reliably learn one- and two-dimensional observables of event data with up to 26 dimensions. The best performance is achieved when true observations are used to construct a prior for sampling the decoder of the B-VAE. Furthermore, the latent space dimensionality should be lower but close to the input dimension of the event. In addition, we show that an offset for sampling the Gaussian distributions in latent space improves the generalization capabilities of the B-VAE.

## Results

We study the behavior of generative models as defined in "Deep Learning Methods" on the three different data sets described in "Monte Carlo Data". Most of the conducted studies focus on the $t\bar{t}$ dataset beginning in "$pp \rightarrow t\bar{t} \rightarrow 4$ jets+1 or 2 leptons" and we only present short, preliminary studies on the two-body and the leptonic Z decay in "Two-body decay toy model" and "e+e− → Z → l+l−". We show several one- and two-dimensional observables for all data sets and evaluate the trained $t\bar{t}$ generator with figures of merit we define in "Figures of Merit". Our study finds that by using the B-VAE, we are able to capture the underlying distribution such that we can generate a collection of events that is in very good agreement with the distributions found in MC event data with 12 times more events than in the training data. Our study finds that many GAN architectures with default parameters and the standard VAE do not perform well. The best GAN results in this study are achieved by the DijetGAN[20] with the implementation as delivered by the authors and the LSGAN[25] with the recommended hyperparameters. The failure of the standard VAE is accounted to the fact that the distributions of encoded physical events in latent space is not a standard normal distribution. We find that the density information buffer can circumvent this issue. To this end we perform a brief parameter scan beyond dim z and B for the smudge factors $\alpha$ and offsets $\gamma$. We also present the performance of the optimized B-VAE. Additionally, we investigate whether improvements to the density information buffer can be achieved by performing a Kernel Density Estimation, creating a Gaussian Mixture Model or learning the latent code density with another VAE. Finally, we perform sanity checks with the $t\bar{t}$ dataset in "Sanity checks", obtain benchmark performances from traditional methods and test whether our proposed method is trivial, i.e., whether it is only learning the identity function.

**Two-body decay toy model**. The comparison of the generative model performances for the toy model in Fig. 1a indicates that the B-VAE with an adjusted prior, given in Eq. (17), is the best investigated ML technique that is able to reliably model the $p_x$, $p_y$, and $p_z$ distributions when compared to regular GANs and VAEs with a standard normal prior, although these models still give good approximations. We find that all models learn the relativistic dispersion relation which underlines the findings in[26,27]. It is noteworthy that for this data set, we only try regular GANs with small capacities and find that they can already model the distributions reasonably well. We confirm the findings in[20,21] that it is problematic for GANs to learn the uniform distribution in $\phi$. While it is one of the few deviations that

occur in[20,21] circumvents the issue with $\phi$ by only learning $\Delta\phi$ between the two jets and manually sampling $\phi_{j_1} \sim U(-\pi, \pi)$. It is questionable whether this technique can be generalized to higher multiplicities.

$e^+e^- \rightarrow Z \rightarrow l^+l^-$. Figures 1 b and 2 show the results for the Z events, where the Z boson decays leptonically. Here we find that the B-VAE is able to accurately generate events that respect the probability distribution of the physical events. We find very good agreement between the B-VAE and physical events for distributions of $p_T$, $\theta$, and $\phi$ and good agreement for invariant mass $M_{inv}$ of the lepton pair around 91 GeV. While the standard VAE fails for the momentum conservation beside having a peak around 0, the B-VAE is much closer to the true distribution. When displaying $\phi$, $\theta$ and the transverse momentum $p_T$ of lepton 1 against lepton 2 (Fig. 2), we find good agreement for the B-VAE, while the standard VAE results in a smeared out distribution. In addition, it can be seen that the events generated by the standard VAE are not always produced back to back but are heavily smeared. We conclude that if we do not use density information buffering, a standard VAE is not able to accurately generate events that follow the Monte Carlo distributions. In particular, events with four leptons are sometimes generated if no buffering is used.

$pp \rightarrow t\bar{t} \rightarrow 4$ jets+1 or 2 leptons. Here we present and discuss the results for the more complicated $t\bar{t}$ production with a subsequent semi-leptonic decay. We train the generative models on events that have four jets and up to two leptons in the final state such that their input and output dimension is 26. For simplicity we do not discriminate between b-jets and light-flavored jets, nor between different kinds of leptons. A jet is defined as a clustered object that has a minimum transverse momentum ($p_T$) of 20 GeV in the Monte Carlo simulation. We first explore the hyperparameter space of the B-VAE in dim z, B, $\alpha$, $\gamma$, and recommend a best practice for the creation of a generative model for physical events. Subsequently we investigate various methods to learn the latent code density of encoded ground truth data. Finally, we try to create a generative model for physical events with several GAN architectures.

Tables 1 and 2 show the top-15 performances of (dim z, B, $\alpha$, $\gamma$) combinations evaluated on the figures of merit defined in "Figures of Merit". For all possible combinations of dim z and B as defined in "Generative Models" we have separately investigated

$$\gamma = \{0.01, 0.05, 0.1\},$$
$$\alpha = \{1, 5, 10\}.$$

For the $\gamma$-study we fixed $\alpha = 1$ and for the $\alpha$-study we fixed $\gamma = 0$. Tables 1 and 2 show the ranking in $\delta_{1D}$ for the studies on $\gamma$ and $\alpha$ respectively.

It is not surprising that the best performance in $\delta_{1D}$ is attained by the B-VAE with the highest latent code dimensionality, the lowest B and $\alpha = 1$, $\gamma = 0$. The downside however is a very poor performance in $\delta_{OF}$. Comparing to the values for the 5% Gaussian smearing of events in Table 3.

They are very similar in $\delta$ but even worse in $\delta_{OF}$ and thus, this model provides no advantage over simple smearing without using machine learning techniques: it essentially learns to reproduce the training data. We observe similar patterns for the ranking in $\delta$: the models that perform best only provide a small advantage. Other models do provide a bigger advantage but there is a trade-off between performance in $\delta$ and $\delta_{OF}$ that can in principle be weighted arbitrarily. By introducing the factor $\alpha$ we smear the B-VAE events in latent space. Models with neither

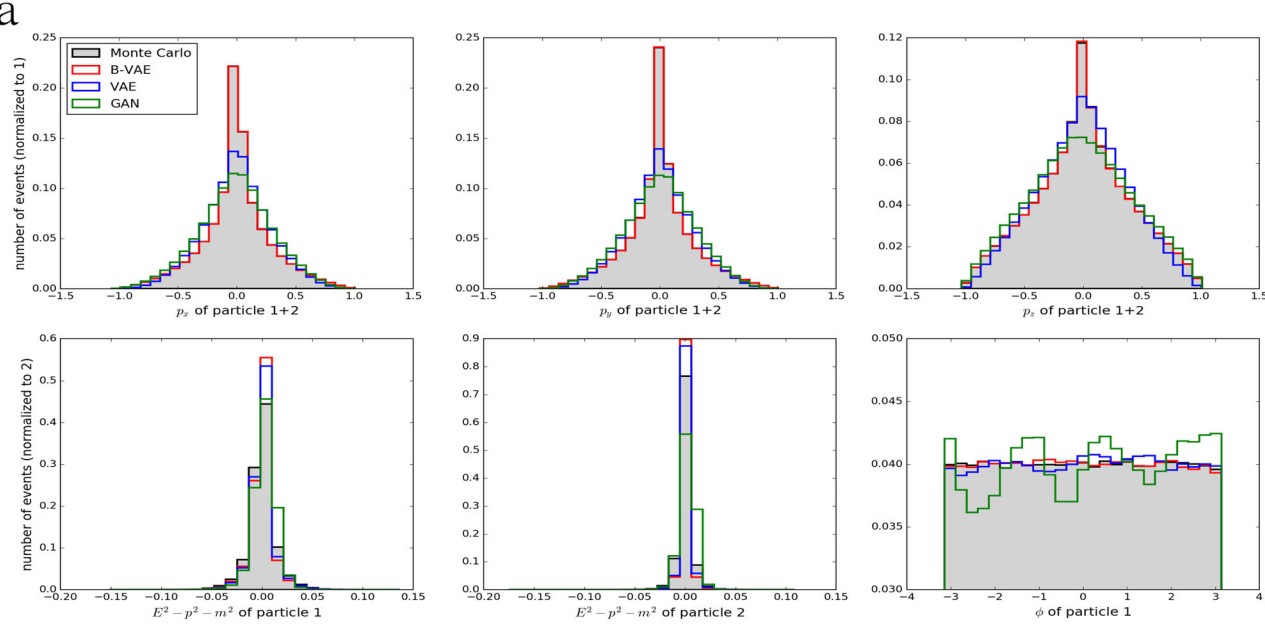

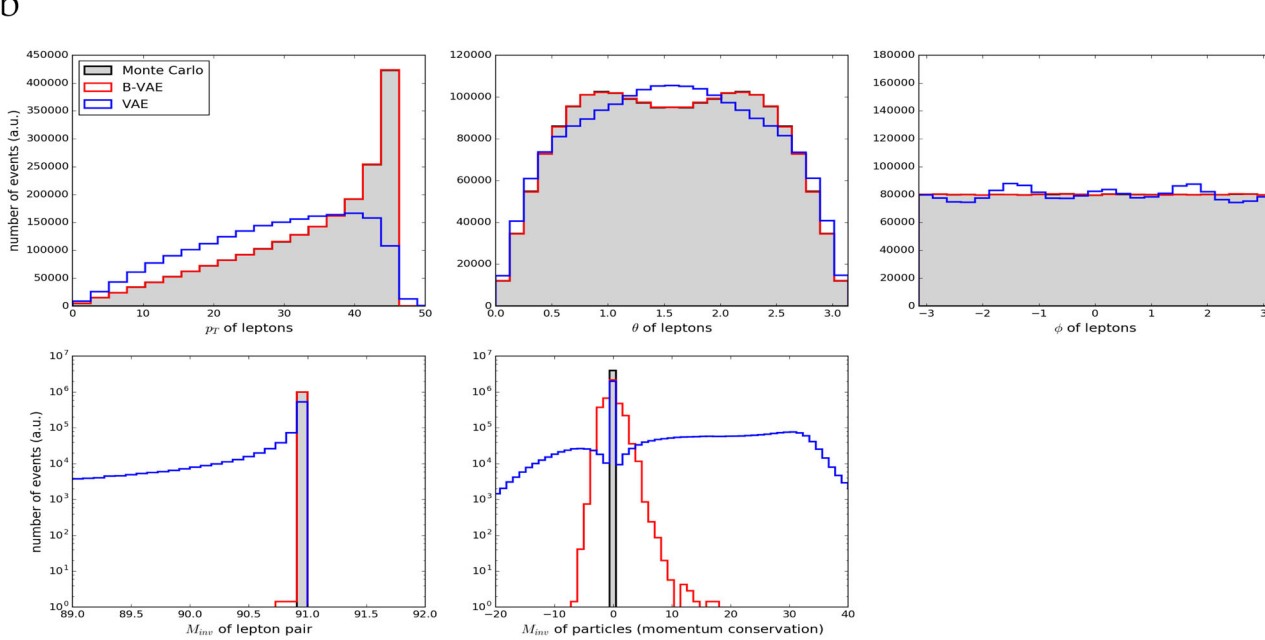

**Fig. 1 Histograms of two-body and leptonic Z decay.** Events that are generated by a Monte Carlo generator (gray) and several machine learning models for a toy two-body decay in **a** and the leptonic Z decay in **b**. Shown are histograms for the VAE with a standard normal prior (blue), the B-VAE with a density information buffer (red) and by the GAN (green, only in **a**)).

smearing nor an offset perform poorly in $\delta_{OF}$, whereas models with $B > 10^{-5}$ perform poorly in $\delta$. For illustrative purposes, we proceed to show and discuss details for the model we consider best: dim $\mathbf{z} = 20, B = 10^{-6}, \alpha = 1, \gamma = 0.05$. Figure 3 shows the comparison between B-VAE events and ground truth data in 29 one-dimensional histograms for this model:

- $E$, $p_T$, $\eta$, and $\phi$ for all four jets and the leading lepton,
- MET and MET$\phi$,
- $\Delta\phi$ between MET and leading lepton,
- $\Delta R$ between leading and subleading jets and

- the invariant mass $M_{\mathrm{inv}}$ for 2, 3, and 4 jets and 4 jets + 1 and 2 leptons.

Note that the training data and the density information buffer consist of the same $10^5$ samples that were used to generate $1.2 \times 10^6$ events which are compared to $1.2 \times 10^6$ ground truth samples. We observe that the ground truth and generated distributions generally are in good agreement. For the invariant masses we again observe deviations in the tail of the distribution. For MET, MET$\phi$, $\Delta\phi$, and $\Delta R$ we see almost perfect agreement.

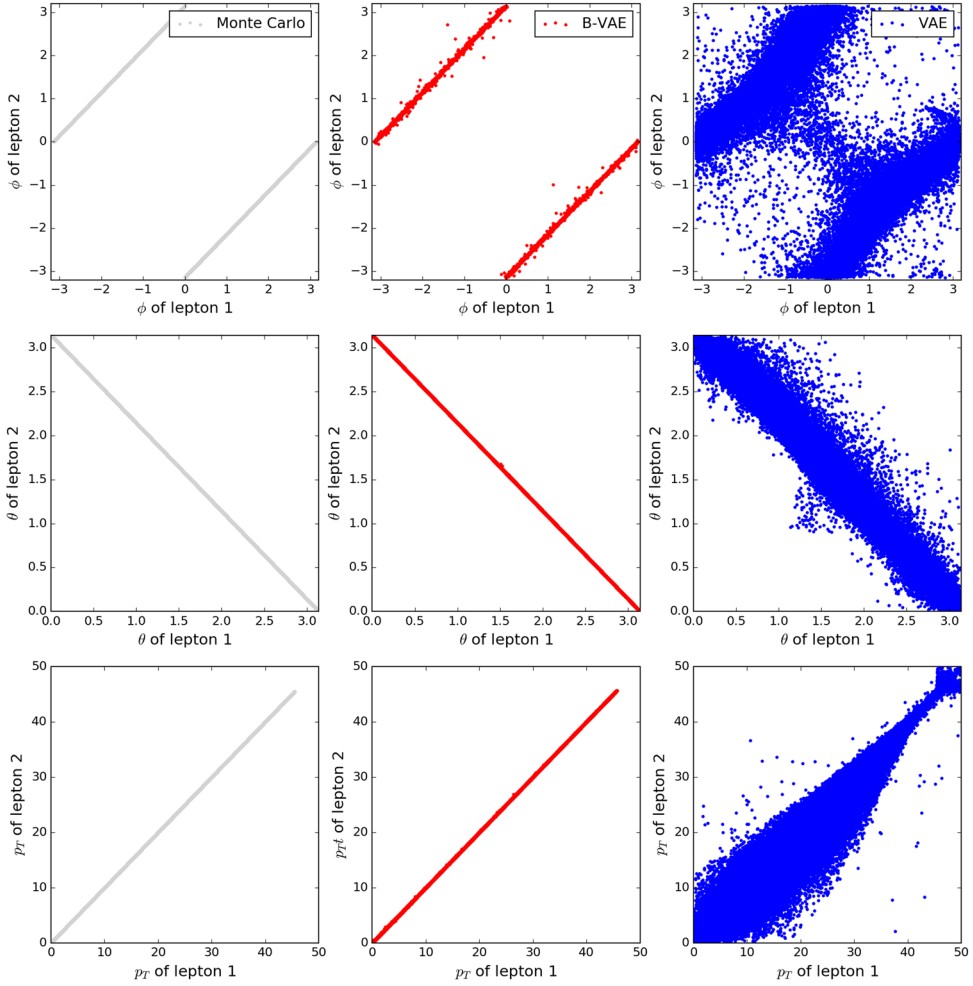

**Fig. 2 2D Histograms for leptonic Z decay events.** Events that are generated by the Monte Carlo generator for the $e^+e^- \to Z \to l^+l^-$ process (gray points), by the VAE with a standard normal prior (blue points) and by the B-VAE with a buffering of density information in the latent space (red points). The top line shows the azimuthal angle $\phi$ for lepton 1 and 2. The middle line shows $\theta$ for lepton 1 and 2. The bottom line shows the $p_T$ of lepton 1 and 2 (in GeV). The variance in the distribution of $\phi$ is an artifact of the simulation used to generate the data, not a statistical fluctuation.

| **Table 1 B-VAE performance with varying $\gamma$.** | | |
|---|---|---|
| **(dim z, B, $\alpha$, $\gamma$)** | **$\delta$** | **$\delta_{OF}$** |
| $(20, 10^{-6}, 1, 0.01)$ | 0.0076 | 3.69 |
| $(20, 10^{-7}, 1, 0.01)$ | 0.0090 | 3.81 |
| $(20, 10^{-6}, 1, 0.05)$ | 0.0090 | 1.01 |
| $(16, 10^{-7}, 1, 0.01)$ | 0.0095 | 4.29 |
| $(16, 10^{-6}, 1, 0.01)$ | 0.0101 | 3.30 |
| $(16, 10^{-6}, 1, 0.05)$ | 0.0122 | 0.51 |
| $(16, 10^{-5}, 1, 0.05)$ | 0.0137 | 0.46 |
| $(24, 10^{-5}, 1, 0.05)$ | 0.0138 | 0.65 |
| $(16, 10^{-5}, 1, 0.01)$ | 0.0148 | 1.23 |
| $(20, 10^{-5}, 1, 0.01)$ | 0.0148 | 1.18 |
| $(24, 10^{-5}, 1, 0.01)$ | 0.0149 | 1.06 |
| $(28, 10^{-7}, 1, 0.01)$ | 0.0155 | 4.22 |
| $(24, 10^{-7}, 1, 0.01)$ | 0.0156 | 3.63 |
| $(24, 10^{-6}, 1, 0.01)$ | 0.0165 | 3.68 |
| $(28, 10^{-6}, 1, 0.01)$ | 0.0176 | 3.71 |

The combinations of dim **z**, $B$, $\alpha = 1$ and $\gamma$ giving the top-15 performance w.r.t. $\delta$ with the corresponding $\delta_{OF}$.

| **Table 2 B-VAE performance with varying $\alpha$.** | | |
|---|---|---|
| **(dim z, B, $\alpha$, $\gamma$)** | **$\delta$** | **$\delta_{OF}$** |
| $(28, 10^{-7}, 1, 0)$ | 0.0066 | 94.00 |
| $(24, 10^{-7}, 1, 0)$ | 0.0074 | 97.92 |
| $(20, 10^{-6}, 1, 0)$ | 0.0075 | 17.48 |
| $(20, 10^{-7}, 1, 0)$ | 0.0084 | 106.64 |
| $(20, 10^{-7}, 5, 0)$ | 0.0088 | 5.67 |
| $(16, 10^{-7}, 5, 0)$ | 0.0093 | 7.96 |
| $(16, 10^{-7}, 1, 0)$ | 0.0094 | 133.48 |
| $(16, 10^{-6}, 1, 0)$ | 0.0102 | 14.02 |
| $(16, 10^{-7}, 10, 0)$ | 0.0102 | 2.05 |
| $(20, 10^{-7}, 10, 0)$ | 0.0112 | 1.65 |
| $(24, 10^{-7}, 5, 0)$ | 0.0144 | 4.55 |
| $(24, 10^{-6}, 1, 0)$ | 0.0156 | 15.45 |
| $(28, 10^{-6}, 1, 0)$ | 0.0162 | 13.52 |
| $(28, 10^{-7}, 5, 0)$ | 0.0166 | 4.60 |
| $(28, 10^{-6}, 5, 0)$ | 0.0189 | 0.68 |

The combinations of dim **z**, $B$, $\alpha$ and $\gamma = 0$ giving the top-15 performance w.r.t. $\delta$ with the corresponding $\delta_{OF}$.

**Table 3 Performance of KDE, GMM and Smearing of events**
**Evaluation of event modeling on figures of merit $\delta$ and $\delta_{OF}$.**

| Model | $\delta$ | $\delta_{OF}$ |
|---|---|---|
| KDE | 0.6038 | 4.99 |
| GMM, 50 | 0.1078 | 16.33 |
| GMM, 100 | 0.0948 | 20.43 |
| GMM, 1000 | 0.0874 | 12.09 |
| 5% Smearing | 0.0093 | 3.53 |
| 10% Smearing | 0.0192 | 3.89 |

Generating $10^7 t\bar{t}$ events with the VAE has taken 177.5 seconds on an Intel i7-4790K and is therefore several orders of magnitude faster than the traditional MC methods.

Figure 4 shows eight histograms of $\phi$ of the leading jet vs. $\phi$ of the next to leading jet ($\phi_1$ vs $\phi_2$) that were created using the B-VAE with $\dim \mathbf{z} = 20, B = 10^{-6}, \alpha = 1, \gamma = 0.05$. The left column shows the histogram for the full range $[-\pi, \pi] \times [-\pi, \pi]$ whereas the right column shows the same histogram zoomed in on $[2, 3] \times [2, 3]$. The first row displays the training data consisting of $10^5$ events. The second and third row of Fig. 4 show $1.2 \times 10^6$ ground truth and B-VAE events respectively allowing for a comparison of how well the B-VAE generalizes considering it was trained on only $10^5$ events. The amount of empty bins (holes) present for the ground truth and B-VAE events is very similar. Also, the general features of the generated distribution are in very good agreement with the ground truth. However, one can spot two shortcomings:

- the presented model smears the detector granularity that is visible in $\phi$ due to the $\gamma$ parameter which would be learned for $\alpha = 1$ and $\gamma = 0$ and
- generator artifacts appear around $(\pm\pi, 0)$ and $(0, \pm\pi)$. For $E$, $p_T$, and $\eta$ we observe larger deviations in the tails of the distributions while for $\phi$ we only observe slightly more events produced around $\pm\pi$.

The first effect is most likely due to the $\gamma$ parameter and the second effect was already expected from the deviations in the one-dimensional azimuthal distributions around $\pm\pi$.

Figure 5 shows how the fraction of empty bins evolves with respect to the number of bins in 2D histograms of $\eta_{j_1}$ vs. $\eta_{j_2}$ and $\phi_{j_1}$ vs. $\phi_{j_2}$ for several models including the ground truth. One can see that our chosen model, whose performance was presented in Figs. 3 and 4, also accurately follows the fraction of empty bins of the Monte Carlo data.

As discussed in "Latent code density estimation" we compare four different methods for constructing a prior for the generative model. We compare a KDE, three GMMs, and several S-VAEs to the explicit latent code density of encoded ground truth data. To demonstrate this we choose the same B-VAE model as in the preceding paragraph: $(20, 10^{-6}, 1, 0.05)$. Figure 6 shows histograms of all 20 latent code dimensions coming from the different approaches. We observe that all dimensions are generally modeled well by all approaches, except for the S-VAEs with extreme values of $B$. This is an expected result since the encoder $q_\phi(\mathbf{z}|\mathbf{x})$ transforms the input into multivariate Gaussians for which a density estimation is much easier than for such non-Gaussian densities present in physical events.

Table 4 shows the performance of the different approaches. It is remarkable that the KDE and GMM models of the prior $p(\mathbf{z})$ provide such good performance in $\delta$, especially the GMM with 1000 components. A drawback for all of the models that try to learn the latent code density is that the resulting performance

in $\delta$ and $\delta_{OF}$ is very poor when compared to the explicit use of the density information buffer.

We compare several state of the art GAN architectures in Table 5.

Table 5 shows the evaluation of the GAN models on our figures of merit. However, we find that no GAN architecture we tried is able to provide a satisfactory performance with respect to $\delta$ and that all of the tried architectures perform worse than traditional methods such as KDE and GMM except for the LSGAN. The best GAN we find is the LSGAN that, in contrast to all GANs we try otherwise, outperforms all traditional and several B-VAE models with respect to $\delta_{OF}$. Figure 7 shows the loss curves for the GAN architectures that are also shown in Table 5 and the B-VAE.

Considering the GAN literature, the results found are not surprising; the authors in[20,21] report difficulties when trying to learn $\phi$. Several other papers report that it is very difficult or technically unfeasible to learn densities with GANs[28–31]. Some of these papers even show that the regular GAN and the WGAN can even fail to learn a combination of 2D Gaussians and that they are not suited to evaluate densities by design[31].

Note that all the GAN models we have tried here were trained using the hyperparameters that were recommended in the corresponding papers. However, each of these models is accompanied by large hyperparameter spaces that impact the performance of the generator. The poor performance we find for most GAN models, therefore, does not imply that GANs are ruled out as potential generative models.

**Sanity checks**. We perform two sanity checks: (1) we show that two traditional density learning algorithms, Kernel Density Estimation and Gaussian Mixture Models, do not work well when applied directly on the events. (2) we check whether the VAE learns the identity function. Both checks are performed on the $t\bar{t}$ data.

We perform a KDE with an initial grid-search as described in "Latent Code Density Estimation" to find the optimal bandwidth on a reduced data set with $10^4$ samples and then perform a KDE with $h_{opt}$ on $10^5$ samples. Additionally, we create a GMM of those $10^5$ samples with 50, 100, and 1000 components with a maximum of 500 iterations. Subsequently, we generate $1.2 \times 10^6$ samples from the KDE and the three GMM models and evaluate them with our figures of merit $\delta$ and $\delta_{OF}$ as presented in Table 3. In addition we take $10^5$ events and smear them by sampling from a Gaussian around these events. To this end, we pre-process them in the same way as above and multiply every dimension of every event with $\mathcal{N}(1, \sigma^2 = \{0.05, 0.1\})$ and sample 12 times per event. Table 3 generally shows poor performance of all models, especially for $\delta_{OF}$. Only the smearing shows good performance for $\delta$ This procedure however does not respect the correlations in the data and therefore also performs poorly for $\delta_{OF}$.

When $\dim \mathbf{z}$ is greater than or equal to the number of dimensions of the training data, it becomes questionable whether a VAE is merely learning the identity function, i.e., whether

$$p_\theta(\tilde{\mathbf{x}}^i | \mathbf{z}(\mathbf{x}^i)) = \delta(\mathbf{x} - \mathbf{x}^i). \tag{1}$$

Since $q_\phi(\mathbf{z}|\mathbf{x}^i)$ always had non-zero variance, no delta functions occur practically. However, one can notice a bias in some variables when feeding random uniform noise $\mathbf{x}_{test} \sim U(0, 1)$ into the VAE. This is no surprise since the encoder and decoder are constructed to learn a function that can reconstruct the input. In Fig. 8 we show the reconstructions for the 26-dimensional $t\bar{t}$ events of a VAE with a 20-dimensional latent space and $B = 10^{-6}$ and the reconstructions of the same VAE for $\mathbf{x}_{test} \sim U(0, 1)$, where we clearly see that the VAE does not simply learn the identity function. The parameters $\alpha, B, \gamma$, and $\dim \mathbf{z}$ allow one to tune how the B-VAE generalizes.

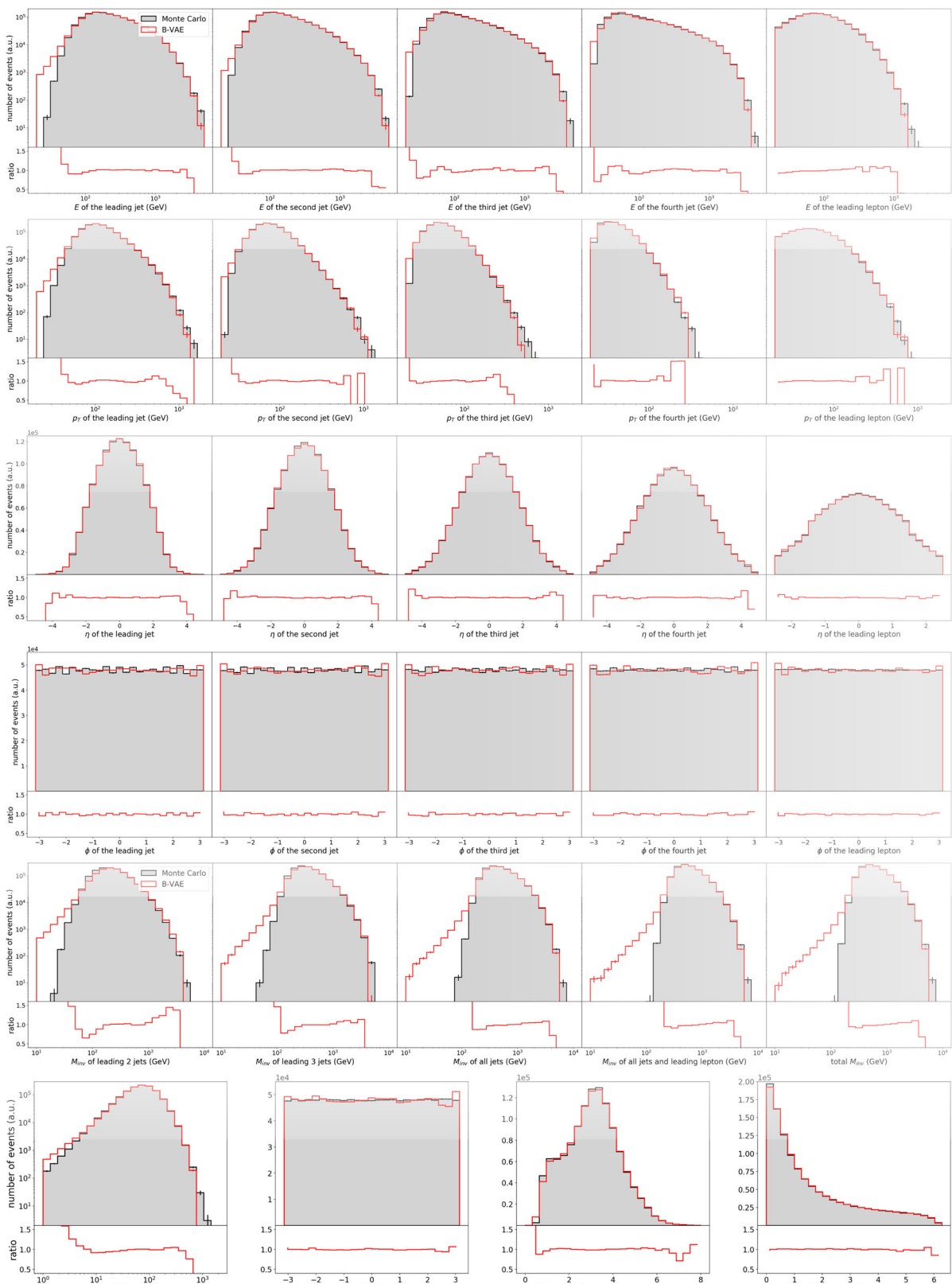

**Fig. 3 1D Histograms of the distributions.** Distributions for the ground truth (gray) and samples generated by the B-VAE with dim $\mathbf{z} = 20$, $B = 10^{-6}$, $\alpha = 1$ and $\gamma = 0.05$ (red).

## Discussion

We have found that the B-VAE as a deep generative model can be a good generator of collision data. In this section, we discuss several further applications of this work such as anomaly detection and improved MC integration. We demonstrate the option of how one can utilize the B-VAE to steer the event generation.

To steer the event generation we need to find out which regions in latent space correspond to which events generated by the

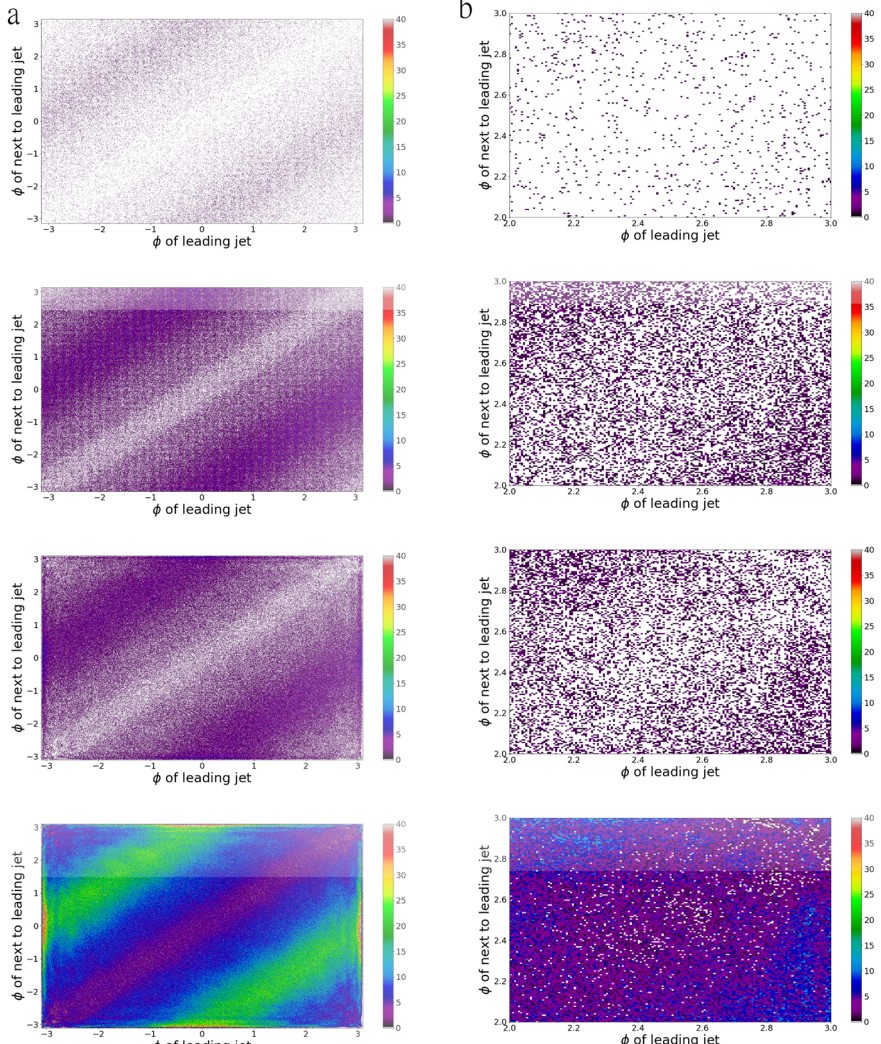

**Fig. 4 2D Histograms of $t\bar{t}$ events for $\phi_{j_1}$ vs. $\phi_{j_2}$.** The first row shows the training data of the B-VAEs: $10^5$ ground truth events. The second row shows $1.2 \times 10^6$ ground truth events. The third row shows $1.2 \times 10^6$ events created by the B-VAE with dim $\mathbf{z} = 20$, $B = 10^{-6}$, $\alpha = 1$, $\gamma = 0.05$. The fourth row shows $10^7$ events generated by the same B-VAE, i.e., the data it generates is 100 times larger than the data it was trained on. In **a** events are showed for the full range of $(\phi_{j_1}, \phi_{j_2})$: $[-\pi, \pi] \times [-\pi, \pi]$ while in **b** a zoom on $(\phi_1, \phi_2) \in [2, 3] \times [2, 3]$ is done. The full range is subdivided into $1000 \times 1000$ bins.

decoder, i.e., we want to find a mapping from relevant latent space volumes to phase space volumes. To this end, we perform a principal component analysis of the latent space representation of physical events. The PCA is an orthogonal transformation of the data that defines new axes such that the first component accounts for most of the variance in the dataset. We look at the first two principal components, sample a grid in these components and apply the inverse PCA transformation to get back to a latent space representation. We choose 64 points in latent space that were picked after finding that physical events in PCA space are distributed on an approximately circular area. Because of that finding we created an equidistant $8 \times 8$ grid in polar coordinates $r$ and $\phi$. The grid in PCA space is then transformed back to a latent space representation and used as input for the decoder to generate events that are being displayed in Fig. 9. The 64 chosen points on a polar grid correspond to the events in Fig. 9. This is effectively a two-dimensional PCA map of latent space. Observing the event displays reveals that we are in fact able to capture where we find events with what number of jets and leptons, what order of MET and what kind of orientations. In case one wants to produce events that e.g., look like event 62, one can do this by sampling around $r = 3.5$ and $\phi = 225°$ in PCA space, then

transform these events back to a latent space representation and to use that as input for the decoder. This will offer the possibility to narrow down the characteristics of the events even further and many iterations of this procedure will finally allow the generation of events with arbitrarily precise characteristics. Alternatively, one could create a classifier that defines boundaries of a latent space volume and corresponds to the desired phase space volume.

Having found that the B-VAE can be used to sample highly complex probability distributions, one possible application may be to provide a very efficient method for the phase space integration of multi-leg matrix elements. Recent work has shown that machine learning approaches to Monte Carlo integration of multidimensional probability distributions[32] and phase space integration of matrix elements[33] may be able to obtain much better rejection efficiency than the current widely used methods[8]. We point out that event weights can be obtained from the B-VAE in similar fashion to the above papers.

The reconstruction of noise and test events in Fig. 8 clearly shows that $t\bar{t}$ events beyond the training data are (a) embedded well in latent space and (b) reconstructed very well when compared to the reconstruction of noise. This suggests that one can use the (relative) reconstruction loss histograms or the (relative)

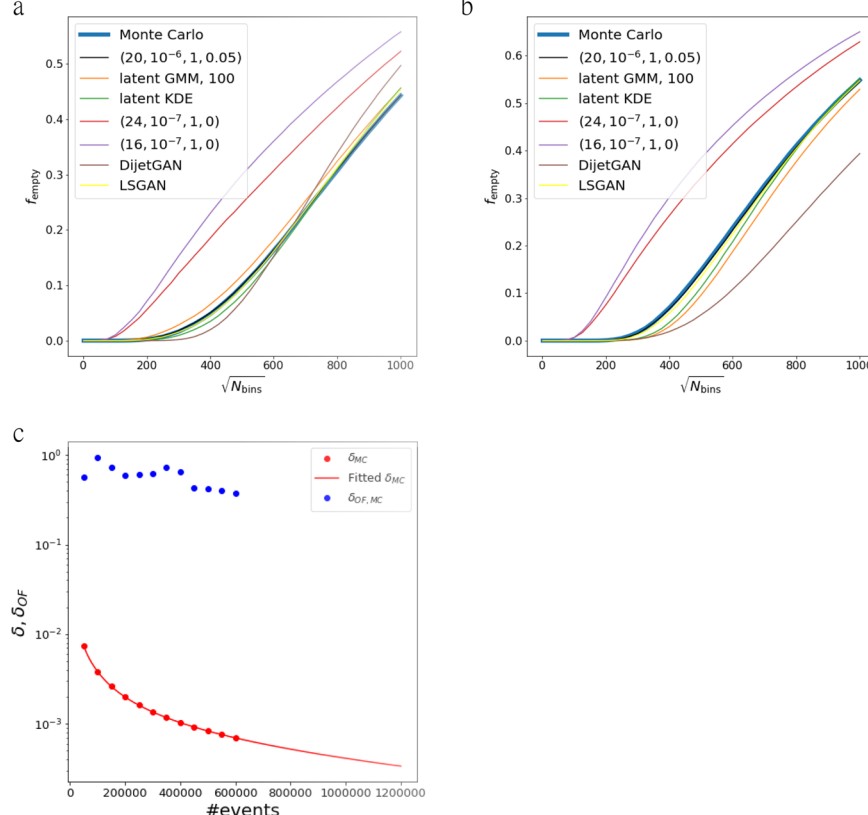

**Fig. 5 $\delta_{OF}$ for several models and Monte Carlo data. a** and **b** show the fraction of empty bins $f_e$ plotted versus $\sqrt{N_{bins}}$ for 1.1 million events from the Monte Carlo data and several models. **c** shows the reality and expectation for $\delta$ and $\delta_{OF}$ for the Monte Carlo data up until $6 \times 10^5$ events in steps of $5 \times 10^4$.

reconstruction losses to detect anomalies, i.e., departures from the training data in terms of single events or their frequency of occurrence. The obvious use case of this is to train a B-VAE on a mixture of standard model events to detect anomalies in experimental data that correspond to new physics similarly to[34]. The B-VAE makes it possible to increase the ability to reconstruct the training and test data compared to a normal VAE, so it may be a better anomaly detector.

We have provided more evidence for the capability of deep generative models to learn physical processes. To compare the performance of all of the investigated models, we have introduced two figures of merit, $\delta$ and $\delta_{OF}$. In particular, we describe and optimize a method for this task: the B-VAE. Several GAN architectures with recommended hyperparameters and the VAE with a standard normal prior fail to correctly produce events with the right frequency of occurrence. By creating a density information buffer with encoded ground truth data we presented a way to generate events whose probabilistic character-istics are in very good agreement with those found in the ground truth data. We identified the relevant hyperparameters of the B-VAE that allow for the optimization of its general-ization properties and performed a first exploration of that hyperparameter space. We find that the dimensionality of the latent space should be smaller than, but close to, the input dimension. We find that it is necessary to heavily weight the reconstruction loss to create an accurate generative model and to tune the underestimated variance of the latent code. We have tested several traditional density estimation methods to learn the latent code density of encoded ground truth data, and concluded that the explicit use of the density information buffer with the parameters $\alpha$ and $\gamma$ performs better. In a final step, we have investigated several GAN architectures with default

hyperparameters but failed to create a model that successfully generates physical events with the right densities. Improve-ments could be made by performing a stricter model selection and to sweep through the full hyperparameter space beyond the hyperparameter recommendations given in the corresponding GAN papers. More generally, the GAN training procedure may be improved because the simultaneous gradient ascent that is currently used to find local Nash equilibria of the two-player game has issues that may be overcome by other objectives like the consensus optimization[28] or by approaches such as the generative adversarial density estimator[31].

By performing a principal component analysis of the latent space representations of MC events and a subsequent exploration of the corresponding PCA space we introduced an option to steer the event generation. In "Results" we demonstrate that the sta-tistics generalize to some degree. In future work it will be necessary to identify to which degree the implicit interpolation of the density $p_\theta(\mathbf{x})$ generalizes beyond the observed ground truth - and to maximize it. Another missing piece to complete the puzzle, is to find which generative models can describe processes that contain both events with very high and low multiplicities with up to twenty or more final state objects. Independent of what the outcome will be, potential improvements to all presented tech-niques can be made by incorporating auxiliary features as in[18,21]. Furthermore, improvements can be made by adding regression terms to the loss function that penalize deviations from the desired distributions in the generated data as in[21] and by utilizing classification terms that force the number of objects and the corresponding object type to be correct. Another promising class of methods to create generative models are flow-based models[35–37] and a thorough comparison of all available meth-ods would be useful.

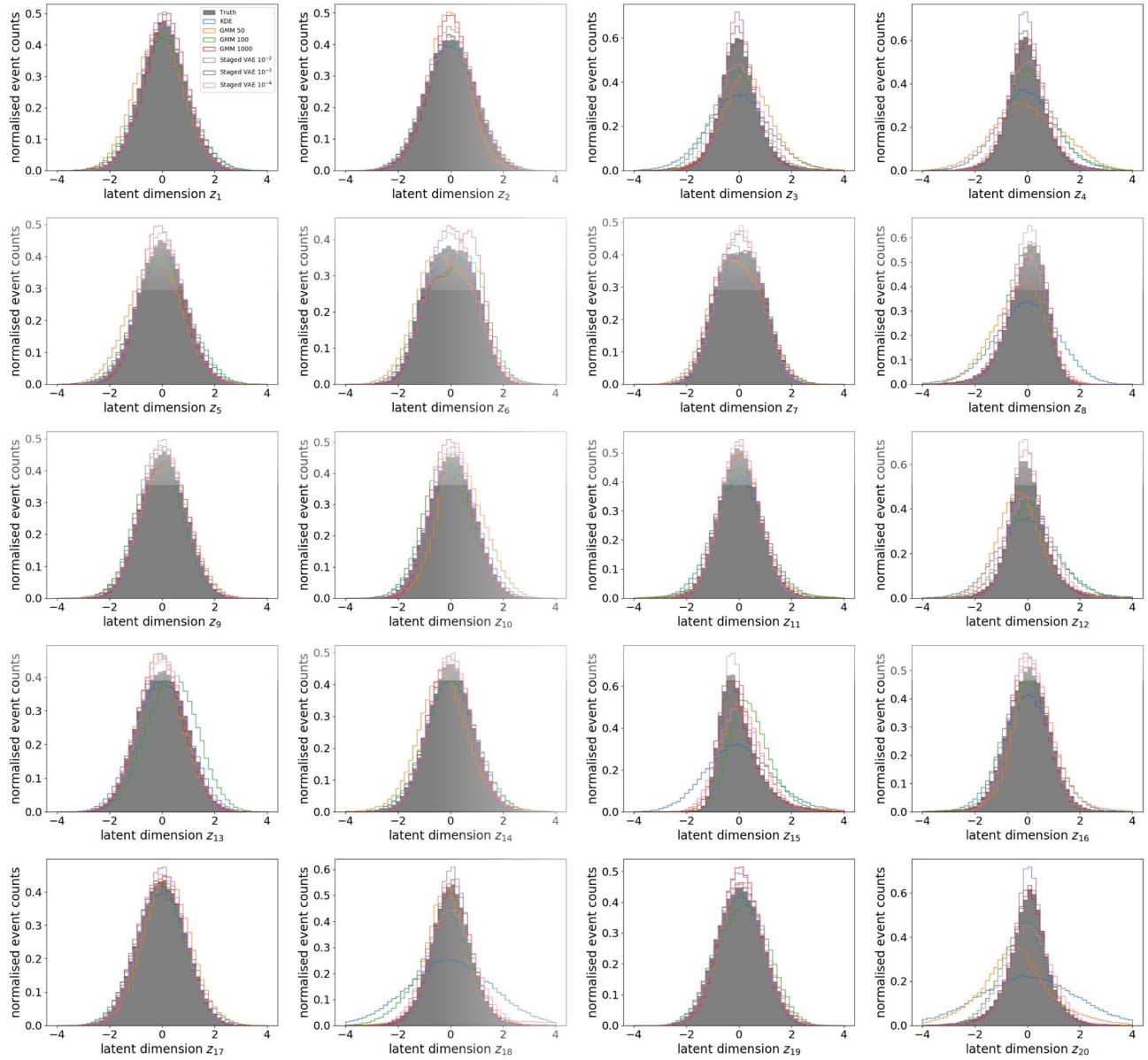

**Fig. 6 Histograms for all latent space dimensions for several models and the ground truth.** The use models are Kernel Density Estimation, Gaussian Mixture Models and Staged Variational Autoencoders. The ground truth itself is an approximation extracted from latent codes given the encoding of $10^5$ MC events by a VAE with dim $\mathbf{z} = 20$ and $B = 10^{-6}$.

### Table 4 Performance with Latent Space Models.

| Model | $\delta$ | $\delta_{\mathrm{OF}}$ |
|---|---|---|
| KDE | 0.2631 | 7.30 |
| GMM, 50 | 0.0297 | 11.54 |
| GMM, 100 | 0.0266 | 12.70 |
| GMM, 1000 | 0.0302 | 7.64 |
| S-VAE, $B = 1$ | 1.4452 | 840.60 |
| S-VAE, $B = 0.1$ | 1.1438 | 756.56 |
| S-VAE, $B = 0.01$ | 0.0689 | 12.73 |
| S-VAE, $B = 10^{-3}$ | 0.0438 | 3.36 |
| S-VAE, $B = 10^{-4}$ | 0.0992 | 9.26 |
| S-VAE, $B = 10^{-5}$ | 0.3179 | 12.33 |
| S-VAE, $B = 10^{-6}$ | 0.8753 | 238.33 |

$\delta$ and $\delta_{\mathrm{OF}}$ of the B-VAE with different latent space density estimation techniques.

### Table 5 GAN Performance.

| GAN model | $\delta$ | $\delta_{\mathrm{OF}}$ |
|---|---|---|
| DijetGAN | 0.3477 | 19.70 |
| LSGAN | 0.3592 | 3.03 |
| MMD-GAN | 0.9454 | 642.20 |
| WGAN-GP | 1.1605 | 723.79 |
| WGAN | 1.0672 | 840.63 |

$\delta$ and $\delta_{\mathrm{OF}}$ for several GAN architectures.

All in all, the results of this investigation indicate usefulness of the hereby proposed method not only for particle physics but for all branches of science that involve computationally expensive Monte Carlo simulations, that have the interest to create a

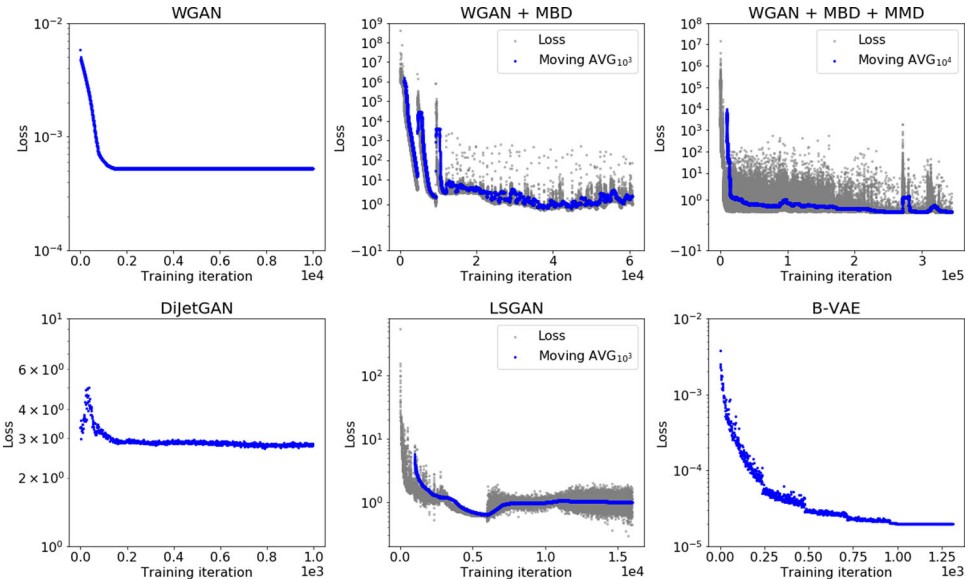

**Fig. 7 Training loss as a function of training step.** The WGAN + MBD, WGAN + MBD + MMD and LSGAN loss behaves chaotically (gray dots), so a moving average as plotted as well to show the average behavior over time. The window size of the moving average is specified in the legend.

**Fig. 8 Input vs. reconstruction.** First four columns of uniform noise $x \sim U(0, 1)$ and last four columns of real events for a VAE with dim $\mathbf{z} = 20$ and $B = 10^{-6}$.

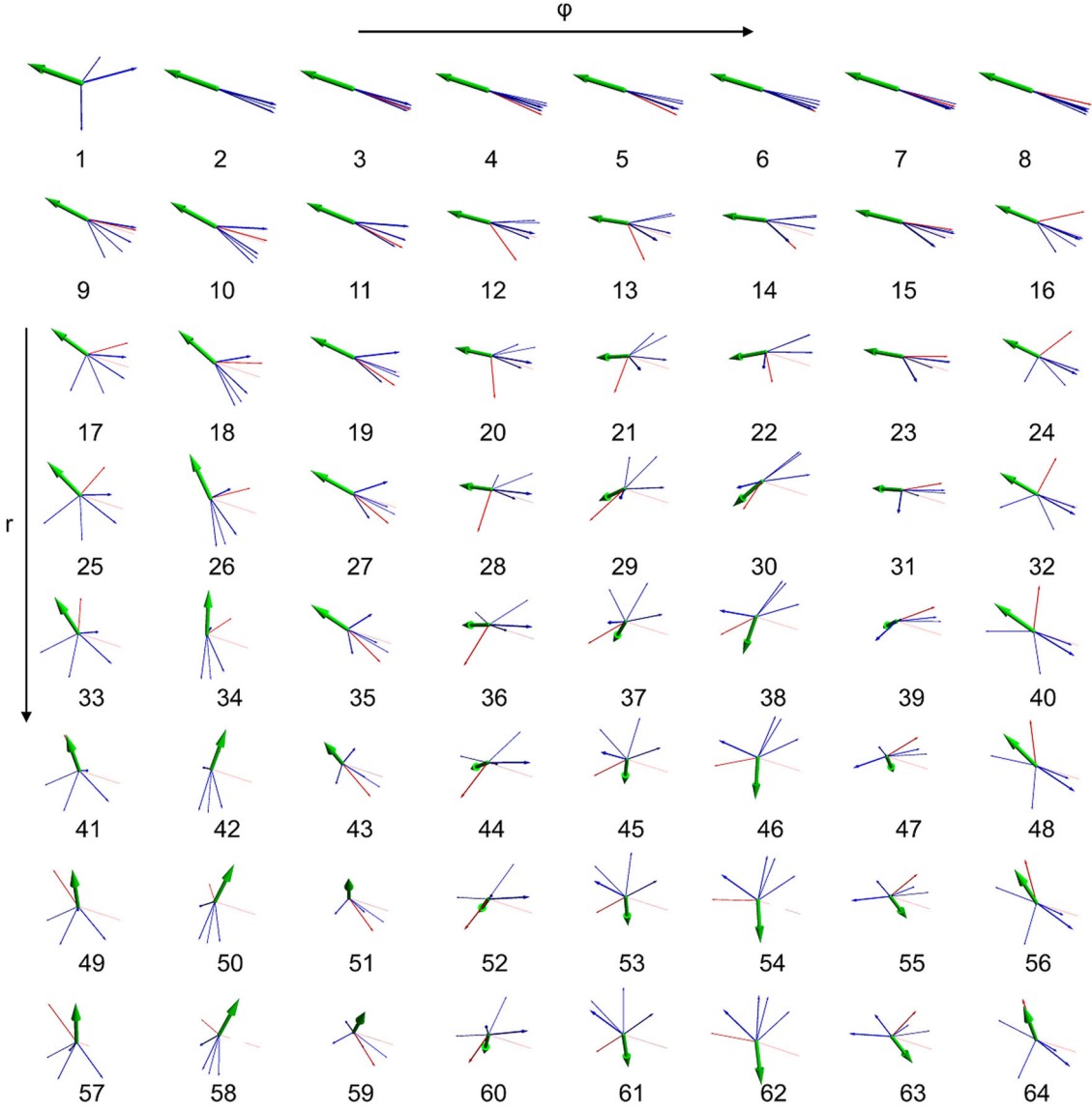

**Fig. 9 Visualization of the first two components of a principal component analysis of encoded Monte Carlo events in latent space.** This shows the events created from a 8 × 8 polar grid in PCA space. These 64 points chosen in PCA space are transformed to a latent space representation and fed into the decoder. The output of the decoder is then visualized: blue arrows indicate jets, red arrows indicate leptons and the green arrow indicates the missing energy vector. The thickness of the arrow corresponds to the relative energy of the 4-vector to the other 4-vectors in the same event. The latent space grid is set up in $(r, \phi)$ coordinates, where steps of 3.4/7 are taken in $r$ with an initial $r = 0.1$, increasing from top to bottom, and steps of 45° are taken in $\phi$, increasing from left to right.

generative model from experimental data, or that have the need to sample from high-dimensional and complex distributions.

## Methods

**Monte Carlo Data**. We study the generation of physical events using three different sets of generated events: 1. a simple toy-model, 2. Z-boson production in $e^+e^-$ collisions and 3. top quark production and decay in proton collisions, i.e., $pp \rightarrow t\bar{t}$. Here, we describe the procedures for obtaining the data sets.

*10-dimensional toy model*. For the toy model we assume a stationary particle with mass $M$ decaying into two particles with masses $m_1$ and $m_2$ and calculate their momentum 4-vectors by sampling $m_1$, $m_2$, $\theta$, and $\phi$ from uniform distributions $10^6$ times. $\theta$ and $\phi$ are the polar and azimuthal angle of the direction into which particle 1 travels:

$$\theta = \arccos \frac{p_z}{\sqrt{p_x^2 + p_y^2 + p_z^2}} \;,\; \phi = \arctan \frac{p_y}{p_x}. \quad (2)$$

These angles and momentum conservation fix the direction of particle 2. The quantities of the model that are used as training data for the generative models are

the energies $E_1$, $E_2$ of the daughter particles, the phase space components $p_x$, $p_y$, $p_z$ for each particle and their masses $m_1$ and $m_2$. This introduces a degeneracy with the goal of checking whether the generative models learn the relativistic dispersion relations

$$E^2 - \mathbf{p}^2 - m^2 = 0. \quad (3)$$

*16-dimensional $e^+e^- \rightarrow Z \rightarrow l^+l^-$.* We generate $10^6$ events of the $e^+e^- \rightarrow Z \rightarrow l^+l^-$ ($l \equiv e, \mu$) process at matrix element level with a center-of-mass energy of 91 GeV using MG5_aMC@NLO v6.3.2[1]. The four-momenta of the produced leptons are extracted from the events given in LHEF format[38], and are directly used as input data for the generative models. The dimensionality of the input and output data is therefore 16: ($E_{e^-}, p_{x,e^-}, p_{y,e^-}, p_{z,e^-}, E_{e^+}, p_{x,e^+}, p_{y,e^+}, p_{z,e^+}, E_{\mu^-}, p_{x,\mu^-}, p_{y,\mu^-}, p_{z,\mu^-}, E_{\mu^+}, p_{x,\mu^+}, p_{y,\mu^+}, p_{z,\mu^+}$) and will always contain eight zeros, since the events consist of $e^+e^-$ or $\mu^+\mu^-$.

*26-dimensional $pp \rightarrow t\bar{t}$.* We generate $1.2 \times 10^6$ events of $pp \rightarrow t\bar{t}$, where at least one of the top-quarks is required to decay leptonically. We used MG5_aMC@NLO v6.3.2[1] for the matrix element generation, using the NNPDF PDF set[39]. Madgraph is interfaced to Pythia 8.2[2], which handles showering and hadronization. The matching with the parton shower is done using the MLM merging prescription[40].

Finally, a quick detector simulation is done with Delphes 3[3,41], using the ATLAS detector card. For all final state objects we use $(E, p_T, \eta, \phi)$ as training data and also include MET and MET$\phi$. We have five or six objects in the final state, four jets, and one or two leptons, i.e., our generative models have a 26 dimensional input and output, while those with only one lepton contain four zeros at the position of the second lepton.

**Deep learning methods.** This section summarizes the methodology used to investigate deep and traditional generative models to produce a realistic collection of events from a physical process. We present the generative techniques we have applied to the data sets, GANs and VAEs, with a focus on our technique: an explicit probabilistic model, the B-VAE, which is a method combining a density information buffer with a variant of the VAE. Subsequently, we discuss several traditional methods to learn the latent code densities and finally, we present figures of merit to assess the performance of the generative models.

*Generative models.* In this section, we give a brief description of GANs and a thorough description of our B-VAE technique. For the latter, we provide the details of the corresponding architecture as well as hyperparameters and training procedures, and also show how the density information buffer is created and how it is utilized to generate events. The GANs and VAEs are trained on an Nvidia Geforce GTX 970 and a Tesla K40m GPU using tensorflow-gpu 1.14.0[42], Keras 2.2.5[43], and cuDNN 7.6.1[44].

*Generative Adversarial Networks.* GANs learn to generate samples from a data distribution by searching for the global Nash equilibrium in a two-player game. The two players are neural networks: one that tries to generate samples that convince the other, a discriminator that tries to distinguish real from fake data. There are many possibilities to realize this, accompanied by large hyperparameter spaces. We try to create event generators with several recent GAN architectures:

- the regular Jensen-Shannon GAN[13] that was only applied to the first toy model dataset,
- several more recent GAN architectures that have been applied to the $t\bar{t}$ dataset: Wasserstein GAN (WGAN)[45], WGAN with Gradient Penalty (WGAN-GP)[46], Least Squares GAN (LSGAN)[25], Maximum Mean Discrepancy GAN (MMDGAN)[47].

We use the recommended hyperparameters from the corresponding papers. Note here that an extensive hyperparameter scan may yield GANs that perform better than those reported in this paper. The performance of the GAN models we present serves as baselines.

*Explicit probabilistic models.* Consider that our data, $N$ particle physics events $\mathbf{X} = \{\mathbf{x}^i\}_{i=1}^N$, are the result of a stochastic process and that this process is not known exactly. It depends on some hidden variables called latent code $\mathbf{z}$. With this in mind one may think of event generation as a two-step process: (1) sampling from a parameterized prior $p_\theta(\mathbf{z})$ (2) sampling $\mathbf{x}^i$ from the conditional distribution $p_\theta(\mathbf{x}^i|\mathbf{z})$, representing the likelihood. For deep neural networks the marginal likelihood

$$p_\theta(\mathbf{x}) = \int p_\theta(\mathbf{z}) p_\theta(\mathbf{x}|\mathbf{z}) d\mathbf{z} \tag{4}$$

is often intractable. Learning the hidden stochastic process that creates physical events from simulated or experimental data requires us to have access to an efficient approximation of the parameters $\boldsymbol{\theta}$. To solve this issue an approximation to the intractable true posterior $p_\theta(\mathbf{z}|\mathbf{x})$ is created: a probabilistic encoder $q_\phi(\mathbf{z}|\mathbf{x})$. Given a data point $x^i$ it will produce a distribution over the latent code $\mathbf{z}$ from which the data point might have been generated. Similarly, a probabilistic decoder $p_\theta(\mathbf{x}|\mathbf{z})$ is introduced that produces a distribution over possible events $\mathbf{x}^i$ given some latent code $\mathbf{z}$. In this approach, the encoder and decoder are deep neural networks whose parameters $\boldsymbol{\phi}$ and $\boldsymbol{\theta}$ are learned jointly.

The marginal likelihood

$$\log p_\theta(\mathbf{x}^1, \ldots, \mathbf{x}^N) = \sum_{i=1}^N \log p_\theta(\mathbf{x}^i) \tag{5}$$

can be written as a sum of the likelihood of individual data points. Using that

$$\log p(\mathbf{x}^i) = \log \mathbb{E}_{p(\mathbf{z}|\mathbf{x}^i)} \left[ \frac{p(\mathbf{x}^i, \mathbf{z})}{p(\mathbf{z}|\mathbf{x}^i)} \right] \tag{6}$$

and applying Jensen's inequality, one finds that

$$\log p(\mathbf{x}^i) \geq \mathbb{E}_{p(\mathbf{z}|\mathbf{x}^i)} \left[ \log \frac{p(\mathbf{x}^i, \mathbf{z})}{p(\mathbf{z}|\mathbf{x}^i)} \right]. \tag{7}$$

For this situation, one must substitute $p(\mathbf{z}|\mathbf{x}^i) \rightarrow q_\phi(\mathbf{z}|\mathbf{x}^i)$ since we don't know the true posterior but have the approximating encoder $q_\phi(\mathbf{z}|\mathbf{x}^i)$. From here one can derive the variational lower bound $\mathcal{L}(\theta, \phi; \mathbf{x}^i)$ and find that

$$\log p_\theta(\mathbf{x}^i) = D_{\mathrm{KL}}(q_\phi(\mathbf{z}|\mathbf{x}^i) \parallel p_\theta(\mathbf{z}|\mathbf{x}^i)) + \mathcal{L}(\theta, \phi; \mathbf{x}^i). \tag{8}$$

$D_{\mathrm{KL}}$ measures the distance between the approximate and the true posterior and since $D_{\mathrm{KL}} \geq 0$, $\mathcal{L}(\theta, \phi; \mathbf{x}^i)$ is called the variational lower bound of the marginal likelihood

$$\begin{aligned} \mathcal{L}(\theta, \phi; \mathbf{x}) &= \mathbb{E}_{q_\phi(\mathbf{z}|\mathbf{x})} \left[ -\log q_\phi(\mathbf{z}|\mathbf{x}) + \log p_\theta(\mathbf{x}, \mathbf{z}) \right] \\ &= -D_{\mathrm{KL}}(q_\phi(\mathbf{z}|\mathbf{x}) \parallel p_\theta(\mathbf{z})) \\ &\quad + \mathbb{E}_{q_\phi(\mathbf{z}|\mathbf{x})} \left[ \log \left( p_\theta(\mathbf{x}|\mathbf{z}) \right) \right]. \end{aligned} \tag{9}$$

We optimize $\mathcal{L}(\theta, \phi; \mathbf{x}^i)$ with respect to its variational and generative parameters $\phi$ and $\theta$.

*Variational Autoencoder.* Using the Auto-Encoding Variational Bayes Algorithm[22] (AEVB) a practical estimator of the lower bound is maximized: for a fixed $q_\phi(\mathbf{z}|\mathbf{x})$ one reparametrizes $\hat{\mathbf{z}} \sim q_\phi(\mathbf{z}|\mathbf{x})$ using a differentiable transformation $g_\phi(\epsilon, \mathbf{x})$, $\epsilon \sim \mathcal{N}(0, 1)$ with an auxiliary noise variable $\epsilon$. Choosing $\mathbf{z} \sim p(\mathbf{z}|\mathbf{x}) = \mathcal{N}(\mu, \sigma^2)$ with a diagonal covariance structure, such that

$$\log q_\phi(\mathbf{z}|\mathbf{x}) = \log \mathcal{N}(\mathbf{z}; \mu, \sigma^2 \mathbb{1}) \tag{10}$$

where $\mu$ and $\sigma^2$ are outputs of the encoding deep neural network. Reparametrizing $\mathbf{z} = \mu + \sigma \odot \epsilon$ yields the Variational Autoencoder (VAE)[22]. In that case the first term in Eq. (9) can be calculated analytically:

$$-D_{\mathrm{KL}}(q_\phi(\mathbf{z}|\mathbf{x}) \parallel p_\theta(\mathbf{z})) = \frac{1}{2} \sum_{j=1}^{\dim \mathbf{z}} 1 + \log(\sigma_j^2) - \mu_j^2 - \sigma_j^2. \tag{11}$$

The second term in Eq. (9) corresponds to the negative reconstruction error that, summed over a batch of samples, is proportional to the mean squared error (MSE) between the input $\mathbf{x}^i$ and its reconstruction given the probabilistic encoder and decoder. The authors in[22] state that for batch-sizes $M > 100$ it is sufficient to sample $\epsilon$ once which is adopted in our implementation. By calculating the lower bound for a batch of $M$ samples $\mathbf{X}^M \subset \mathbf{X}$ they construct the estimator of $\mathcal{L}$:

$$\mathcal{L}(\theta, \phi; \mathbf{X}) \simeq \mathcal{L}^M(\theta, \phi; \mathbf{X}^M) = \frac{N}{M} \sum_{i=1}^M \tilde{\mathcal{L}}(\theta, \phi; \mathbf{x}^i), \tag{12}$$

where

$$\begin{aligned} \tilde{\mathcal{L}}(\theta, \phi; \mathbf{x}^i) &= -D_{\mathrm{KL}}(q_\phi(\mathbf{z}|\mathbf{x}^i) \parallel p_\theta(\mathbf{z})) \\ &\quad + \log(p_\theta(\mathbf{x}^i | g_\phi(\epsilon^i, \mathbf{x}^i))). \end{aligned} \tag{13}$$

We use the gradients $\nabla_{\theta,\phi} \mathcal{L}^M(\theta, \phi; \mathbf{X}^M, \epsilon)$ for the SWATS optimization procedure[48], beginning the training with the Adam optimizer[49] and switching to stochastic gradient descent. Practically, the maximization of the lower bound is turned into the minimization of the positive $D_{\mathrm{KL}}$ and the MSE such that the loss function of the VAE $L \propto D_{\mathrm{KL}} + MSE$. In our approach, we introduce a multiplicative factor $B$ to tune the relative importance of both terms. The authors in[50] introduce a similar factor $\beta$, but their goal is to disentangle the latent code by choosing $\beta > 1$ such that each dimension is more closely related to features of the output. In contrast, we choose $B \ll 1$ to emphasize a good reconstruction. The loss function of the VAE can subsequently be written as

$$L = \frac{1}{M} \sum_{i=1}^M (1 - B) \cdot MSE + B \cdot D_{\mathrm{KL}}. \tag{14}$$

This however also implies that $D_{\mathrm{KL}}(q_\phi(\mathbf{z}|\mathbf{x}) \parallel p_\theta(\mathbf{z}))$ is less important, i.e., there is a much smaller penalty when the latent code distribution deviates from a standard Gaussian. This incentivizes narrower Gaussians because the mean squared error for a single event reconstruction grows as the sampling of the Gaussians in latent space occur further from the mean. Note that for $B = 0$ one obtains the same loss function as for a standard autoencoder[51]: the reconstruction error between input and output. Although the standard deviations will be small, the VAE will still maintain its explicit probabilistic character because it contains probabilistic nodes whose outputs are taken to be the mean and logarithmic variance, while the standard autoencoder does not.

The encoders and the decoders of our VAEs have the same architectures consisting of four (toy model and $Z \rightarrow l^+l^-$) or six hidden layers ($pp \rightarrow t\bar{t}$) with 128 neurons each and shortcut connections between every other layer[52,53]. We choose $B = 3 \cdot 10^{-6}$ for the toy model and $Z \rightarrow l^+l^-$. The number of latent space dimensions is nine for the toy model and 10 for $Z \rightarrow l^+l^-$. For the toy model we use a simple training procedure using the Adam optimizer with default values for 100 epochs. For $Z \rightarrow l^+l^-$ we employ a learning rate scheduling for $7 \times 80$ epochs and SWATS[48], i.e., switching from Adam to SGD during training.

For $pp \rightarrow t\bar{t}$ we perform a scan over hyperparameters with

$$\dim \mathbf{z} = \{16, 20, 24, 28\},$$
$$B = \{10^{-7}, 10^{-6}, 10^{-5}, 10^{-4}\},$$

We perform this scan on a small training data set with $10^5$ samples. We use a batch size of 1024 and the exponential linear unit[54] as the activation function of hidden layers. The output layer of the decoder is a hyperbolic tangent such that we need to pre- and postprocess the input and output of the VAE. We do this by dividing each dimension of the input by the maximum of absolute values found in the training

data. We apply this pre- and post-processing in all cases. We initialize the hidden layers following a normal distribution with mean 0 and a variance of $(1.55/128)^{0.5}$ such that the variance of the initial weights is approximately equal to the variance after applying the activation function on the weights[54]. For $pp \rightarrow t\bar{t}$ the setup is identical except for the number of epochs: we train $4 \times 240$ epochs with Adam and then for $4 \times 120$ epochs with SGD. Due to the increasing complexity of the data sets we perform more thorough training procedures.

*Latent Code Density Estimation.* In the case of VAEs the prior $p(\mathbf{z}) = \mathcal{N}(0, 1)$ used to sample $p_\theta(\mathbf{x}|\mathbf{z})$ is not identical to the distribution over the latent code $\mathbf{z}$ resulting from the encoding of true observations $q_\phi(\mathbf{z}|\mathbf{X})$. The generated distribution over $\mathbf{x}$ given $p_\theta(\mathbf{x}|\mathbf{z})$ therefore doesn't match the reference when assuming a unit Gaussian over $\mathbf{z}$. We address this issue by estimating the prior $p(\mathbf{z})$ for the probabilistic decoder from data using a strategy similar to the Empirical Bayes method[55]. We collect observations $\mathbf{Z} = \{\mathbf{z}_1, ..., \mathbf{z}_m\}$ by sampling $q_\phi(\mathbf{z}|\mathbf{X}_L)$ where $\mathbf{X}_L \subset \mathbf{X}$ is a subset of physical events. $\mathbf{Z}$ is then used as the data for another density estimation to create a generative model for $p(\mathbf{z})$: this is what we call the density information buffer. This buffer is used in several ways to construct $p(\mathbf{z})$: we apply Kernel Density Estimation[56], Gaussian Mixture Models using the expectation-maximization algorithm[57], train a staged VAE[58] and directly use the density information buffer. Note that the Kernel Density Estimation and the Gaussian Mixture Models are also used in another context, namely in the attempt to construct such a traditional generative model that is optimized on physical events instead of the latent code as suggested here.

*Kernel Density Estimation.* Given N samples from an unknown density $p$ the kernel density estimator $\hat{p}$ for a point $y$ is constructed via

$$\hat{p}(y) = \sum_{i=1}^{N} K\left(\frac{y - x_i}{h}\right) \tag{15}$$

where the bandwidth $h$ is a smoothing parameter that controls the trade-off between bias and variance. Our experiments make use of $N = \{10^4, 10^5\}$, a Gaussian kernel

$$K(x; h) \propto \exp\left(-\frac{x^2}{2h^2}\right), \tag{16}$$

and have optimized $h$. We use the KDE implementation of scikit-learn[59] that offers a simple way to use the KDE as a generative model and optimize the bandwidth $h$ using GridSearchCV and a fivefold cross validation for 20 samples for $h$ distributed uniformly on a log-scale between 0.1 and 10.

*Gaussian Mixture Models.* Since the VAE also minimizes $D_{KL}(q_\phi(\mathbf{z}|\mathbf{x}) \parallel \mathcal{N}(0, 1))$ it's incentivized that even with low values of $\beta$ the latent code density $q_\phi(\mathbf{z}|\mathbf{X}_L)$ is similar to a Gaussian. It therefore appears promising that a probabilistic model that assumes a finite set of Gaussians with unknown parameters can model the latent code density very well. We use the Gaussian Mixture Model as implemented in scikit-learn, choosing the number of components to be $\{50, 100, 1000\}$ with the full covariance matrix and $10^5$ encodings $\mathbf{z}_i \in \mathbf{Z}$.

*Two-stage VAE.* The idea of the two-stage VAE (S-VAE) is to create another probabilistic decoder $p_\eta(\mathbf{z}|\mathbf{z}')$ from latent code observations $\mathbf{Z}$ that is sampled using $p(\mathbf{z}') = \mathcal{N}(0, 1)$[58]. We use a lower neural capacity for this VAE with three hidden layers with 64 neurons each without shortcut connections for each neural network, and use $B = \{10^{-6}, 10^{-5}, ..., 1\}$. We slightly modify the loss function from Eq. (14) and remove the $(1 - B)$ in front of the MSE term because we want to test higher values of $B$ of up to $B = 1$ and do not want to completely neglect the MSE. Every other hyperparameter including the training procedure is identical to those in the VAE for $pp \rightarrow t\bar{t}$. It is straightforward to expand this even further and also apply KDE, or create a density information buffer from the latent codes $\mathbf{z}'$ to then sample $p_\eta(\mathbf{z}|\mathbf{z}')$ with the data-driven prior.

*Density Information Buffer.* Another way to take care of the mismatch between $p(\mathbf{z})$ and $q_\phi(\mathbf{z})$ is to explicitly construct a prior $p_{\phi, \mathbf{X}_L}(\mathbf{z})$ by aggregating (a subset of) the encodings of the training data:

$$p_{\phi, \mathbf{X}_L}(\mathbf{z}) = \sum_{i=1}^{m} q_\phi(\mathbf{z}|\mathbf{x}^i) p(\mathbf{x}^i) \text{ with } p(\mathbf{x}^i) = \frac{1}{m}. \tag{17}$$

Practically this is done by saving all $\mu^i$ and $\sigma^{2,i}$ for all $m$ events in $\mathbf{X}_L$ to a file, constituting the buffer. The advantage of this procedure is that the correlations are explicitly conserved by construction for the density information buffer while the KDE, GMM, and the staged VAE may only learn an approximation of the correlations in $\mathbf{z} \sim q_\phi(\mathbf{z}|\mathbf{X}_L)$. A disadvantage of this approach is that the resulting density is biased towards the training data, in the sense that the aggregated prior is conditioned on true observations of the latent code for the training data and has a very low variance when $B$ in Eq. (14) is small. One can interpret this as overfitting to the data with respect to the learned density. To counter this effect, we introduce a smudge factor $\alpha$ such that we sample $\mathbf{z}^i \sim \mathcal{N}(\mu^i, \alpha\sigma^{2,i}) \, \forall \, \mathbf{x}^i \in \mathbf{X}_L$. In our experiments we investigate $\alpha = \{1, 5, 10\}$ and only apply $\alpha$ if $\sigma < \sigma_T = 0.05$,

such that

$$\mathbf{z}^i \sim \begin{cases} \mathcal{N}(\mu^i, \alpha\sigma^{2,i}) & \text{if } \sigma < \sigma_T, \\ \mathcal{N}(\mu^i, \sigma^{2,i}) & \text{else} \end{cases} \tag{18}$$

with $\mu^i$ and $\sigma^{2,i}$ being the Gaussian parameters in latent space corresponding to events $i = 1, ..., m$ in $\mathbf{X}_L$. It is straightforward to expand this approach to have more freedom in $\alpha$, e.g., by optimizing $(\alpha_j)_{j=1}^{\dim \mathbf{z}}$, a smudge factor for each latent code dimension. One can include more hyperparameters that can be optimized with respect to figures of merit. By introducing a learnable offset $\gamma_j$ for the standard deviation such that

$$\left(z^i\right)_{j=1}^{\dim \mathbf{z}} \sim \left(\mathcal{N}\left(\mu_j^i, \alpha_j \sigma_j^{2,i} + \gamma_j\right)\right)_{j=1}^{\dim \mathbf{z}}, \tag{19}$$

we have $2 \cdot \dim \mathbf{z}$ additional hyperparameters. More generally we can try to learn a vector-valued function $\boldsymbol{\gamma}(\rho(\mathbf{z}))$ that determines the offset depending on the local point density in latent space. While all of these approaches may allow a generative model to be optimized, it introduces a trade-off by requiring an additional optimization step that increases in complexity with increasing degrees of freedom. In our experiments we only require $\gamma = \gamma_j = \{0.01, 0.05, 0.1\}$ to be the minimal standard deviation, such that

$$\left(z^i\right)_{j=1}^{\dim \mathbf{z}} \sim \left(\mathcal{N}\left(\mu_j^i, \sigma_j^{2,i} + \gamma\right)\right)_{j=1}^{\dim \mathbf{z}}. \tag{20}$$

**Figures of merit.** Having discussed a number of candidate generative models, we now define a method of ranking these models based on their ability to reproduce the densities encoded in the training data. While work that was done so far predominantly relies on $\chi^2$ between observable distributions and pair-wise correlations[20,21], we aim to capture the generative performance more generally. Starting from a total of $1.2 \times 10^6$ Monte Carlo samples, $10^5$ of those samples are used as training data for the generative models. We then produce sets of $1.2 \times 10^6$ events with every model and compare to the Monte Carlo data.

The comparison is carried out by first defining a number of commonly used phenomenological observables. They are the MET, MET$\phi$, and $E$, $p_T$, $\eta$, and $\phi$ of all particles, the two-, three- and four jet, four jet plus one and two lepton mass, the angular distance between the leading and subleading jet

$$\Delta R = \sqrt{(\Delta\phi)^2 + (\Delta\eta)^2} \tag{21}$$

and the azimuthal distance between the leading lepton and the MET. An object is considered to be leading if it has the highest energy in its object class. All possible 2D histograms of these 33 observables are then set up for all models, and are compared with those of the Monte Carlo data. We create 2D histograms with $N_{2D,bins} = 25^2$. We then define

$$\delta = \frac{1}{N_{hist}} \sum_i \sum_{j<i} \chi_{ij}^2 \tag{22}$$

with $i$ and $j$ summing over observables and $N_{hist} = 528$ which represent averages over all histograms of the test statistic

$$\chi^2 = \sum_{u=1}^{N_{bins}} \frac{(p_u - p_u^{MC})^2}{p_u + p_u^{MC}} \tag{23}$$

where $p_u$ and $p_u^{MC}$ are the normalized bin contents of bins $u$[60]. We have verified that the Kullback–Leibler divergence and the Wasserstein distance lead to identical conclusions. This figure of merit is set up to measure correlated performance in bivariate distributions.

A second figure of merit is included with the goal of measuring the rate of overfitting on the training data. If any amount of overfitting occurs, the generative model is expected to not properly populate regions of phase space that are uncovered by the training data. However, given a large enough set of training data, these regions may be small and hard to identify. To test for such a phenomenon, we produce 2D histograms in $\eta_{j_1}$ vs. $\eta_{j_2}$, $\eta_i \in [-2.5, 2.5]$ with a variable number of bins $N_{bins}$ and measure the fraction of empty bins $f_e\left(\sqrt{N_{bins}}\right)$. As $N_{bins}$ is increased, the histogram granularity probes the rate of overfitting in increasingly smaller regions of the phase space. The function $f_e\left(\sqrt{N_{bins}}\right)$ also depends on the underlying probability distribution, and is thus only meaningful in comparison to the Monte Carlo equivalent $f_e^{MC}\left(\sqrt{N_{bins}}\right)$. We, therefore, compute the figure of merit as

$$\delta_{OF} = \int_0^{1000} d\sqrt{N_{bins}} \left| f_e\left(\sqrt{N_{bins}}\right) - f_e^{MC}\left(\sqrt{N_{bins}}\right) \right|. \tag{24}$$

After searching the best model with respect to $\eta_{j_1}$ vs. $\eta_{j_2}$, we independently check its behavior in $\delta_{OF}(\phi_{j_1}, \phi_{j_2})$. The performance, indicated by both figures of merit, is better the lower the value is.

We derive our expectations for the values of $\delta$ and $\delta_{OF}$ for $1.2 \times 10^6$ events by evaluating these figures of merit on MC vs. MC data. Since we only have 1.2 million events in total, we can only evaluate the figures of merit for up to $6 \times 10^5$ events and then extrapolate as shown in Fig. ??. We expect that an ideal generator has $\delta \approx 0.0003$ and $\delta_{OF} \approx 0.5$ for $1.2 \times 10^6$ events. For $\delta$, this value is obtained by fitting the

parameters $A, B, C, D$ to the empirically motivated function

$$\delta(N, A, B, C, D) = \frac{A}{\log(BN + C)} + D \qquad (25)$$

for values of $\delta$ obtained by evaluating the MC dataset with $N = \{50,000, 100,000, \ldots, 600,000\}$ and extrapolating to $N = 120,0000$ using the *curve_fit* function in scipy. The expected $\delta_{OF}$ is assumed to be roughly flat around 0.5. We select our best model by requiring $\delta_{OF} \lesssim 1$ while minimizing $\delta$.

## Data availability

The $t\bar{t}$ dataset that was used to obtain the results in "Sanity checks" and "$pp \rightarrow t\bar{t} \rightarrow 4$ jets+1 or 2 leptons" is available under the https://doi.org/10.5281/zenodo.3560661, https://zenodo.org/record/3560661#.XeaiVehKiUk. All other data that were used to train or that were generated with one of the trained generative models are available from the corresponding author upon request.

## Code availability

The custom code that was created during the work that led to the main results of this article is published in a public GitHub repository: https://github.com/SydneyOtten/DeepEvents.

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

## Acknowledgements

This work was partly funded by and carried out in the SURF Open Innovation Lab project "Machine learning enhanced high-performance computing applications and computations" and was partly performed using the Dutch national e-infrastructure. S.C. and S. O. thank the support by the Netherlands eScience Center under the project iDark: The intelligent Dark Matter Survey. M.v.B. and R.V. acknowledge support by the Foundation for Fundamental Research of Matter (FOM), program 156, "Higgs as Probe and Portal". R.R.d.A. thanks the support from the European Union's Horizon 2020 research and innovation program under the Marie Skłodowska-Curie grant agreement No 674896, the "SOM Sabor y origen de la Materia" MEC projects and the Spanish MINECO Centro de Excelencia Severo Ochoa del IFIC program under grant SEV-2014-0398.

## Author contributions

S.O. contributed to the idea, wrote most of the paper, invented the B-VAE and performed the majority of trainings and the analysis, as well as the data creation for all figures except Fig. 6 and the creation of Figs. 4, 5, and 7. S.C. contributed to the idea, invented the B-VAE, and discussed every section and, also intermediate, results in great detail. W.d.S. contributed to the data generation of the toy model and the initial GAN for the toy model. M.v.B. contributed the $t\bar{t}$ data and Figs. 1, 2, and 9, as well as the discussion and editing of the initial preprint. L.H. contributed to the idea, discussion, and editing with his machine learning expertise, Fig. 6, as well as the training of several GANs. C.v.L. and D.P. contributed to the HPC processing of the data and to discussions. R.R.d.A. contributed the Z-decay data and to discussions of the manuscript. R.V. contributed discussions and implementations of the figures of merit as well as discussions and editing of the manuscript, in particular, "Figures of merit" and Figs. 3 and 8.

## Competing interests

The authors declare no competing interests.
