## [Peer Review File · Nature Communications]

Reviewers' comments:

Reviewer #1 (Remarks to the Author):

Three major claims are reported in this work for which the evidence is partially not sufficient or their motivation is not clearly established.

Firstly, a statement that appears at different locations in this work is that Generative Adversarial Models (GANs) are found to be inferior for the task of modelling physics processes. While this is certainly an interesting hypothesis because GANs have originally been applied to image data, the presented experiment does not provide enough evidence to support its general validity. The study ignores recent, but in the community well-established, progress in the training of generative adversarial networks, for example through the usage of the Wasserstein-distance [1][2]. Furthermore, the network architectures employed in the Variational Autoencoders (VAEs) appear to have more potential than those used for the experiment with GANs. As a result, the authors do not present a fair comparison between VAEs and GANs.

A potential novelty is presented with the second claim. The principal problem with VAEs is the assumption of underlying normal distributions for the latent space variables. This problem is recognised by the authors and is addressed by explicitly injecting the distribution of latent space variables into the decoder. This way, the event generation is significantly improved with respect to traditional VAEs that deploy gaussian priors in latent space.

However, a critical discussion of this method's weaknesses is missing. Only briefly it is mentioned that for small smudge factors, this method tends to "overfit" the data and fails to generalise the generation process. The advantage of estimating the priors for the latent space variables like suggested in equation (5) over traditional kernel density estimation techniques is not clear. Possible correlations of latent space variables seem to be neglected and it is surprising (or a lucky coincidence) that this deficit does not reflect onto the ultimate observable distributions.

Potential further criticism could be the necessity to compute the prior distributions from a sum of many gaussians, each one being attributed with two configurable parameters that all need to be saved eventually. More drastically, one could argue that one might as well compute and sample from the distributions of the final observables for event generation, skipping the latent space and decoder completely. For uncorrelated observables, this would be a valid approach.

In summary, the authors should extend the suggestion of their density buffering by a critical reflection of its implications to event simulation and should provide a comparison to other kernel density estimation techniques for the latent space.

The last claim is the insight that latent space variables and final observables are linked. Hence, one may inject certain points in latent space into the decoder to construct an expected output. But, the usefulness of this method in particle physics simulation is not clear and remains to be motivated by the authors. In addition, the authors fail to provide a generic recipe of their PCA-based method rather than demonstrating it one one example only.

Three experimental datasets have been constructed. They are used sequentially in the argumentation line of this work. Nonetheless, the individual studies on each dataset are linked to

different network architecture- and training tunings aggravating the reproducibility of the results. The comprehension would be enhanced, if the findings were illustrated on only one, e.g. the ttbar, dataset.

The possible applications mentioned in the second half of section IV are weakly motivated and rather vague. In addition, the application of the B-VAE on the CMS open dataset is redundant because its format and content is similar to the ttbar sample. Furthermore, other generative modelling studies prior to the publication of this work have already made use of CMS open data [3].

In general, the language level in this work is rather poor and needs to be improved. Some passages are imprecise, contradictory or vaguely formulated that overall inhibits the understanding of its principal messages. More detailed comments and questions to the authors concerning each passage are attached to this review.

In summary, the submitted manuscript at its current form does not justify its publication. Rather, a major revision focussing on the second claim may be considered for re-submission.

Hereby, the authors are advised to either present more evidence for the insufficient capabilities of GANs in generating physics data or to limit themselves to a brief note on their unsuccessful attempts with GANs.

The potential novelty of the density information buffering needs more distinction from existing methods e.g. by comparing the performance to other results such as [4]. A critical reflection of its weaknesses and technical implications needs to be added.

For this purpose, the investigation might as well focus on only one dataset with one clearly defined network architecture and training strategy to allow for reproducibility of these results.

[1] <https://arxiv.org/abs/1701.07875>

[2] Erdmann, M., Glombitza, J. & Quast, T. *Comput Softw Big Sci* (2019) 3: 4. <https://doi.org/10.1007/s41781-018-0019-7>

[3] Musella, P. & Pandolfi, F. *Comput Softw Big Sci* (2018) 2: 8. <https://doi.org/10.1007/s41781-018-0015-y>

[4] Cerri, O., Nguyen, T.Q., Pierini, M. et al. *J. High Energ. Phys.* (2019) 2019: 36. [https://doi.org/10.1007/JHEP05\(2019\)036](https://doi.org/10.1007/JHEP05(2019)036)

Reviewer #2 (Remarks to the Author):

The paper proposes and studies a variant of the variational autoencoder (VAE) deep generative model for generating physical event data without performing expensive Monte Carlo simulations. The key methodological innovation is to use an informative "prior" for the latent variables which depends on the training data (a form of empirical Bayes), and on the encoder model.

The motivation for the work, accelerating data generation processes for physics applications, is very

strong and could potentially lead to substantial practical impact, by facilitating scientific progress in physics. Very impressive run-time speedups were reported in the results. This advantage is complemented by the capability of the proposed methodology to visualize and steer the event generation process.

The overall approach is interesting and potentially very useful, although I have some concerns with the present manuscript, discussed below.

Note that the data generated from the deep generative models will generally lose some fidelity to the underlying physical process compared to Monte Carlo simulations, although the computational gains will likely make the approach attractive in many cases. The paper could be improved by discussing the limitations and trade-offs of the approach versus Monte Carlo simulations, and likely use cases due to the trade-offs, e.g. for prototyping methods or generating candidate scientific hypotheses before investing in a more expensive simulation. One such limitation is that minimizing the "forward KL-divergence", as performed by VAEs, tends to underestimate the variance in the learned distribution (cf. the textbook by Bishop, "Pattern Recognition and Machine Learning," Figure 10.2 and surrounding text). There should also be a discussion of the possible failure modes of the method.

Regarding the proposed method, while the proposed "density information buffer" is intuitively sensible (it is essentially a semi-parametric estimate of $p(z)$ which replaces an uninformed prior), the paper needs to do more to justify the resulting algorithm's validity. In the manuscript, the notation $p()$ is used to describe the decoder's distribution, but is also used for the encoder's distribution, which is normally denoted by $q()$. This sweeps a slightly unusual assumption under the rug in Equation 5. In Kingma and Welling (2013)'s notation, Equation (5) would be written as:

$$p_{\theta}(z) = \sum_{i=1}^N q_{\phi}(z|x_i) p_{\text{data}}(x_i) .$$

We then clearly see the slightly unusual assumption: $p_{\theta}(z)$ becomes instead $p_{\{\phi, \text{data}\}}(z)$, a function of the variational approximation to the posterior $q_{\phi}(z|x)$ and empirical data distribution $p_{\text{data}}(x)$. Thus, the model (decoder p) and the variational approximation to the model (encoder q) share some parameters. This is non-standard, and I am not sure whether the learning algorithm is still valid. Updating variational parameters ϕ for the encoder q would alter the prior $p(z)$ for the decoder p , which would impact the ELBO objective in an unintended way. Does the overall algorithm still correctly optimize the variational lower bound?

It is also worth considering the impact of Equation 5 on the model. Effectively, the D_{KL} term tries to make $q_{\phi}(z|x_i)$ for each data point similar to the overall $q_{\phi}(z)$ in the dataset, which is an average of $q_{\phi}(z|x_i)$ over all of the x_i 's. This means that the regularizer βD_{KL} is allowed to drift with the estimates in each iteration, instead of penalizing the difference to a fixed standard Gaussian which does not depend on the data. The likely impact is that the VAE model is allowed to fit to the data more closely. The regularization is thus weakened, which could improve the model's fit in big-data scenarios, or cause overfitting in small-data scenarios.

Similarly, it seems conceivable that overfitting the "prior" to the data could lead to degenerate

results. Is the alpha "smudge factor" sufficient to ensure that this is reliably prevented?

Using a data-dependent prior also seems to conflict with the motivating goal of the β -VAE, to learn disentangled representations due to the information bottleneck.

The use of a "prior" $p(z)$ that depends on the entire dataset seems prohibitively expensive in large-scale settings. In this case, how do the authors propose that this cost is managed, e.g. a fixed or random subset of the data points is used instead?

In the paper, a number of experimental results are reported for testing the method's use for statistical physics. The qualitative results regarding goodness of fit appear compelling (note: I do not have the expertise to critique the finer points of physics applications). However, quantitative results on the models' relative performance are not provided, and these are also important. There should be results using some kind of metric of performance to allow for proper rigorous comparison between the methods, which ideally rewards fitting while penalizing over-fitting. E.g., the models' ability to predict held-out data (perhaps using a kernel density estimate constructed on the generated samples?)

An important missing baseline, which needs to be compared to, is the (improved) Wasserstein GAN (WGAN). This method fixes a lot of the issues that were raised regarding GANs, especially the concern that "D would often quickly get too smart."

Arjovsky, M., Chintala, S., & Bottou, L. (2017). Wasserstein GAN. arXiv preprint arXiv:1701.07875.

Gulrajani, I., Ahmed, F., Arjovsky, M., Dumoulin, V., & Courville, A. C. (2017). Improved training of Wasserstein GANs. In Advances in neural information processing systems (pp. 5767-5777).

Minor points:

Pg 1, throughout: The use of "big Oh" $O()$ notation in the paper does not align with the formal mathematical definition of $O()$. E.g., $O(10)$ technically means "in constant time", and not "approximately 10 minutes."

Pg 2, II.A. Define θ , ϕ .

"two-player game minimizes ..." -> "the resulting one-player game minimizes"?

Equation 6: Shouldn't \exp and $\sqrt{\quad}$ be swapped? $\text{Exp}(\text{Enc}[1])$ is the variance (per dimension?), so then the square root of that would be the standard deviation.

Pg 3, define MSE, D_{KL} . Mention that the use of MSE here follows from the Gaussian assumption, but more generally this is log probability.

Define "density information bottleneck".

"gaussian" -> "Gaussian"

Pg 6, "we find very good agreement" - how do you measure agreement?

Reviewer #3 (Remarks to the Author):

I recommend the publication of this paper, provided significant modifications of the text are done. The proposed technique Buffered density VAE is original in the field, and the possible applications in the field are multiple. Hence it meets the originality criterion, and will definitely influence the thinking in the field.

Main modification : the authors are presenting results using GAN, VAE and their proposed modified B-VAE. The results they get with GAN are poor, and the authors claim that GAN cannot work in their case. This is a very strong, unscientific statement, disproven by other papers on similar cases like Musella and Gandolfi arXiv:1805.00850 or Di Sipio et al arXiv 1903.02433. The only statement they can make is that they did not manage themselves. The statement (page 4 left) "Since the discriminator evaluates single events or small batches of events (with minibatch discrimination) in the training process, it is typically non-optimal to find differences in the densities of the training data and the generated data." is plain wrong (if true GANs would never work, but they do). Hence I strongly require that the GAN part of the paper be altogether removed. And then put B-VAE clearly in the title.

General suggestion

The tone of the paper is sometimes aggressive and should be toned down.

Specific suggestions

I

[1-8] : not sure that 4 should be cited just there

[0-13] : not sure that 13 should be cited there

Page 2 left column

"We find that only by using a density" : there is no way to prove this is true. Just say "We find that by using a density"

"We introduce a degeneracy" : please spell it out

Page 2 Right column

b

Please list unambiguously all the variables to be learned (like you did for (a))

c

Same comment

Page 3 left column

Equation 4 MSE and DKL should be indicated two sentences earlier

Page 3 right column

“Large N is favorable” => what value of N is used ?

Page 4 right column

“the standard VAE” (several times)=> “the VAE” (not sure “standard VAE” has any meaning)

“However, we must note that the corresponding widths” => where is this shown ???

Fig 5 and 6 and 10 : indicate in the caption which parameter tuning is the best.

Page 5 left column

“increased diversity among events “ how do you see this ? is this not seen in Fig 7 introduced late ?

10^8 faster : please remove (also from the abstract). Rather only says sthg like :to be compared to up to 10-100s per event.

Fig 7 : would the right column not be more clear with contours ??? Top line : change the color rule, there is no reason the MC to be blue because of less events.

“for the first time”. Remove ! already done in <https://arxiv.org/pdf/1903.02433.pdf>

Page 5 right column

The whole paragraph “ An application of GAN.... Supersymmetry and Z” : sorry I just do not understand. Remove.

Conclusion

“for the first time” => remove!

The value and impact of the paper would be significantly increased if the code was released.

All figures: make sure the unit of the axis is always indicated

Bibliography :

Add Musella and Gandolfi arXiv:1805.00850 and Di Sipio et al arXiv 1903.02433

Dear referees,

Thank you all very much for your comments. They have triggered deeper investigations and led to a major revision of the text.

Attached you can find our answers to the referees in the same order as presented to us. First, I cite reviewers and then answer in arrows.

At the end of this document you can find a list of changes.

Reviewer #1 (Remarks to the Author):

“Three major claims are reported in this work for which the evidence is partially not sufficient or their motivation is not clearly established.

Firstly, a statement that appears at different locations in this work is that Generative Adversarial Models (GANs) are found to be inferior for the task of modelling physics processes. While this is certainly an interesting hypothesis because GANs have originally been applied to image data, the presented experiment does not provide enough evidence to support its general validity. The study ignores recent, but in the community well-established, progress in the training of generative adversarial networks, for example through the usage of the Wasserstein-distance [1][2].

Furthermore, the network architectures employed in the Variational Autoencoders (VAEs) appear to have more potential than those used for the experiment with GANs. As a result, the authors do not present a fair comparison between VAEs and GANs.”

- ➔ We agree and have provided comparisons to more recent and much larger GAN architectures.
- ➔ We do no longer claim that GANs are inferior, just that it is very hard to find a functioning model and that we failed to do so.

“A potential novelty is presented with the second claim. The principal problem with VAEs is the assumption of underlying normal distributions for the latent space variables. This problem is recognised by the authors and is addressed by explicitly injecting the distribution of latent space variables into the decoder. This way, the event generation is significantly improved with respect to traditional VAEs that deploy gaussian priors in latent space.

However, a critical discussion of this method’s weaknesses is missing. Only briefly it is mentioned that for small smudge factors, this method tends to “overfit” the data and fails to generalise the generation process. The advantage of estimating the priors for the latent space variables like suggested in equation (5) over traditional kernel density estimation techniques is not clear. Possible correlations of latent space variables seem to be neglected and it is surprising (or a lucky coincidence) that this deficit does not reflect onto the ultimate observable distributions.

Potential further criticism could be the necessity to compute the prior distributions from a sum of many gaussians, each one being attributed with two configurable parameters that all need to be saved eventually. More drastically, one could argue that one might as well compute and sample from the distributions of the final observables for event generation, skipping the latent space and decoder completely. For uncorrelated observables, this would be a valid approach. In summary, the authors should extend the suggestion of their density buffering by a critical

reflection of its implications to event simulation and should provide a comparison to other kernel density estimation techniques for the latent space.”

- Critical discussion of weaknesses is added to the article.
- Comparison to traditional density estimation techniques in latent space is provided including KDE. We added Gaussian Mixture Models and the staged VAE.
- Comparison to smearing, KDE and GMM performed or trained on the final observables is provided.
- The correlation of latent space variables is conserved in the density information buffer! We have clarified this in the manuscript.

“The last claim is the insight that latent space variables and final observables are linked. Hence, one may inject certain points in latent space into the decoder to construct an expected output. But, the usefulness of this method in particle physics simulation is not clear and remains to be motivated by the authors. In addition, the authors fail to provide a generic recipe of their PCA-based method rather than demonstrating it on one example only.”

- Looking at Figure 9 we think it is obvious that the PCA space of the latent code is linked to the final observables and we have demonstrated the option that this is a handle to steer the event generation.

“Three experimental datasets have been constructed. They are used sequentially in the argumentation line of this work. Nonetheless, the individual studies on each dataset are linked to different network architecture- and training tunings aggravating the reproducibility of the results.

The comprehension would be enhanced, if the findings were illustrated on only one, e.g. the ttbar, dataset.”

- We have significantly expanded and focused on the studies with the ttbar data set. We still included short versions for the other two data sets. From data set 1 we conclude that GANs and VAEs are generally able to learn physical processes. From the study on data set 2 we conclude that standard VAEs do not work well enough and the main studies are performed on ttbar. Regarding the reproducibility we have created a public GitHub repository that contains the B-VAE and GAN models for the ttbar dataset including pretrained weights and the possibility to retrain.

“The possible applications mentioned in the second half of section IV are weakly motivated and rather vague. In addition, the application of the B-VAE on the CMS open dataset is redundant because its format and content is similar to the ttbar sample. Furthermore, other generative modelling studies prior to the publication of this work have already made use of CMS open data [3].”

- We agree with the sentiment regarding the second half of section IV and did a major revision of that paragraph. We still kept the CMS open data example but framed it differently and cited [3].

“In general, the language level in this work is rather poor and needs to be improved. Some passages are imprecise, contradictory or vaguely formulated that overall inhibits the understanding of its principal messages. More detailed comments and questions to the authors concerning each passage are attached to this review.”

- We did our best to improve language and clarity. In several parts of the paper. Attached to the individual responses we present a comprehensive overview of the changes.

“In summary, the submitted manuscript at its current form does not justify its publication. Rather, a major revision focussing on the second claim may be considered for re-submission. Hereby, the authors are advised to either present more evidence for the insufficient capabilities of GANs in generating physics data or to limit themselves to a brief note on their unsuccessful attempts with GANs.”

- We have created a major revision focusing on the second claim. We have provided a broader study on GANs, compare to two contributions considering GANs for event generation and still limit ourselves to a brief note that only our tested GAN models were unsuccessful and that it is a hard problem to create a functioning GAN model. We also briefly present contributions from the computer science literature discussing fundamental issues with GANs and how they may be resolved in future work.

“The potential novelty of the density information buffering needs more distinction from existing methods e.g. by comparing the performance to other results such as [4]. A critical reflection of its weaknesses and and technical implications needs to be added. For this purpose, the investigation might as well focus on only one dataset with one clearly defined network architecture and training strategy to allow for reproducibility of these results.”

- We have provided more distinction by comparing to more models, traditional models such as Kernel Density Estimation and Gaussian Mixture Models, as well as Deep Learning models as provided with the DijetGAN. Additionally we have studied several possibilities to learn the latent code density and thereby provide a variety of options to construct a density information buffer.

[1] <https://arxiv.org/abs/1701.07875>

[2] Erdmann, M., Glombitza, J. & Quast, T. Comput Softw Big Sci (2019) 3: 4.
<https://doi.org/10.1007/s41781-018-0019-7>

[3] Musella, P. & Pandolfi, F. Comput Softw Big Sci (2018) 2: 8.
<https://doi.org/10.1007/s41781-018-0015-y>

[4] Cerri, O., Nguyen, T.Q., Pierini, M. et al. J. High Energ. Phys. (2019) 2019: 36.
[https://doi.org/10.1007/JHEP05\(2019\)036](https://doi.org/10.1007/JHEP05(2019)036)

- All of the new literature is now cited.

Reviewer #2 (Remarks to the Author):

“The paper proposes and studies a variant of the variational autoencoder (VAE) deep generative model for generating physical event data without performing expensive Monte Carlo simulations. The key methodological innovation is to use an informative "prior" for the latent variables which depends on the training data (a form of empirical Bayes), and on the encoder model.

The motivation for the work, accelerating data generation processes for physics applications, is very strong and could potentially lead to substantial practical impact, by facilitating scientific progress in physics. Very impressive run-time speedups were reported in the results. This advantage is complemented by the capability of the proposed methodology to visualize and steer the event generation process.

The overall approach is interesting and potentially very useful, although I have some concerns with the present manuscript, discussed below.”

→ We agree with the summary above

“Note that the data generated from the deep generative models will generally lose some fidelity to the underlying physical process compared to Monte Carlo simulations, although the computational gains will likely make the approach attractive in many cases. The paper could be improved by discussing the limitations and trade-offs of the approach versus Monte Carlo simulations, and likely use cases due to the trade-offs, e.g. for prototyping methods or generating candidate scientific hypotheses before investing in a more expensive simulation. One such limitation is that minimizing the "forward KL-divergence", as performed by VAEs, tends to underestimate the variance in the learned distribution (cf. the textbook by Bishop, "Pattern Recognition and Machine Learning," Figure 10.2 and surrounding text). There should also be a discussion of the possible failure modes of the method.”

→ Departures from the MC simulation are discussed more clearly

→ Failure modes are identified and means to prevent their occurrence investigated

“Regarding the proposed method, while the proposed "density information buffer" is intuitively sensible (it is essentially a semi-parametric estimate of $p(z)$ which replaces an uninformed prior), the paper needs to do more to justify the resulting algorithm's validity. In the manuscript, the notation $p()$ is used to describe the decoder's distribution, but is also used for the encoder's distribution, which is normally denoted by $q()$. This sweeps a slightly unusual assumption under the rug in Equation 5. In Kingma and Welling (2013)'s notation, Equation (5) would be written as:

$$p_{\theta}(z) = \sum_{i=1}^N q_{\phi}(z|x_i) p_{\text{data}}(x_i) .$$

We then clearly see the slightly unusual assumption: $p_{\theta}(z)$ becomes instead $p_{\{\phi, \text{data}\}}(z)$, a function of the variational approximation to the posterior $q_{\phi}(z|x)$ and empirical data distribution $p_{\text{data}}(x)$.”

→ We have added a thorough and proper technical discussion that – hopefully - does no longer sweep anything under the rug.

“Thus, the model (decoder p) and the variational approximation to the model (encoder q) share some parameters. This is non-standard, and I am not sure whether the learning algorithm is still valid. Updating variational parameters ϕ for the encoder q would alter the prior $p(z)$ for the decoder p , which would impact the ELBO objective in an unintended way. Does the overall algorithm still correctly optimize the variational lower bound?”

- It is true that for the generation of new events we sample from a prior $p(z(q))$. However, the ELBO objective is not changed by this during training. Only after training we aggregate encoded ground truth data to construct a prior. Looking at the ELBO after training, we also find that for our prior the first term $D_{\text{KL}}(q(z|x)||p(z))$ should vanish and the reconstruction should also be optimal since training data is reconstructed.

“It is also worth considering the impact of Equation 5 on the model. Effectively, the D_{KL} term tries to make $q_{\phi}(z|x_i)$ for each data point similar to the overall $q_{\phi}(z)$ in the dataset, which is an average of $q_{\phi}(z|x_i)$ over all of the x_i 's. This means that the regularizer βD_{KL} is allowed to drift with the estimates in each iteration, instead of penalizing the difference to a fixed standard Gaussian which does not depend on the data. The likely impact is that the VAE model is allowed to fit to the data more closely. The regularization is thus weakened, which could improve the model's fit in big-data scenarios, or cause overfitting in small-data scenarios.

Similarly, it seems conceivable that overfitting the "prior" to the data could lead to degenerate results. Is the alpha "smudge factor" sufficient to ensure that this is reliably prevented?”

- We have discussed the emphasis of the reconstruction and the potential to overfit in small-data scenarios (using the CMS open data example)
- As discussed in the conclusion of the new article, the smudge factor alpha is not always sufficient to ensure that the model generalizes correctly beyond the training data. We have however investigated ways, discussed options beyond that to optimize the hyperparameters of the B-VAE and showed how well a model trained on 10^5 events generalizes by comparing $1.2 \cdot 10^6$ events generated by the VAE to $1.2 \cdot 10^6$ ground truth events.

“Using a data-dependent prior also seems to conflict with the motivating goal of the β -VAE, to learn disentangled representations due to the information bottleneck.”

- That is true and we have now clearly distinguished our approach from the beta-VAE.

“The use of a "prior" $p(z)$ that depends on the entire dataset seems prohibitively expensive in large-scale settings. In this case, how do the authors propose that this cost is managed, e.g. a fixed or random subset of the data points is used instead?”

- It makes sense to use as many events as possible for constructing the prior. However, a random subset also provides a good approximation. As explained above, the hyperparameters of the B-VAE also depend on the number of training samples / events in the buffer. We have explicitly addressed this issue in the manuscript.

“In the paper, a number of experimental results are reported for testing the method's use for statistical physics. The qualitative results regarding goodness of fit appear compelling (note: I

do not have the expertise to critique the finer points of physics applications). However, quantitative results on the models' relative performance are not provided, and these are also important. There should be results using some kind of metric of performance to allow for proper rigorous comparison between the methods, which ideally rewards fitting while penalizing overfitting. E.g., the models' ability to predict held-out data (perhaps using a kernel density estimate constructed on the generated samples?)”

→ We have introduced figures of merit that also try to penalize overfitting (by penalizing the difference in the fraction of empty bins). Additionally, we have compared the reconstruction of noise input and test data in Figure 1 showing that the B-VAE can predict held-out data nicely.

“An important missing baseline, which needs to be compared to, is the (improved) Wasserstein GAN (WGAN). This method fixes a lot of the issues that were raised regarding GANs, especially the concern that "D would often quickly get too smart."

→ Agreed! We have now included a setup of the WGAN and the WGAN-GP.

“Arjovsky, M., Chintala, S., & Bottou, L. (2017). Wasserstein GAN. arXiv preprint arXiv:1701.07875.

Gulrajani, I., Ahmed, F., Arjovsky, M., Dumoulin, V., & Courville, A. C. (2017). Improved training of Wasserstein GANs. In Advances in neural information processing systems (pp. 5767-5777).”

→ Cited both papers and presented + tested the models

Minor points:

“Pg 1, throughout: The use of "big Oh" $O()$ notation in the paper does not align with the formal mathematical definition of $O()$. E.g., $O(10)$ technically means "in constant time", and not "approximately 10 minutes.””

→ Thank you, corrected!

“Pg 2, II.A. Define θ , ϕ .”

→ Done!

"two-player game minimizes ..." -> "the resulting one-player game minimizes"?

→ Indeed! Adjusted.

“Equation 6: Shouldn't \exp and $\sqrt{\cdot}$ be swapped? $\text{Exp}(\text{Enc}[1])$ is the variance (per dimension?), so then the square root of that would be the standard deviation.

Pg 3, define MSE, D_{KL} . Mention that the use of MSE here follows from the Gaussian assumption, but more generally this is log probability.”

→ We have created an entirely new and much deeper presentation of the theoretical foundations of the Variational Autoencoder and hope that both issues are now presented in a clear way.

“Define "density information bottleneck".”

- Created an own paragraph on the density information buffer and have provided a major revision on the paragraph in which the “density information bottleneck” was mentioned.

"gaussian" -> "Gaussian"

- “Gaussian used everywhere now.”

“Pg 6, "we find very good agreement" - how do you measure agreement?”

- We have introduced figures of merit.

Reviewer #3 (Remarks to the Author):

“I recommend the publication of this paper, provided significant modifications of the text are done.

The proposed technique Buffered density VAE is original in the field, and the possible applications in the field are multiple. Hence it meets the originality criterion, and will definitely influence the thinking in the field.”

→ Thank you!

Main modification : the authors are presenting results using GAN, VAE and their proposed modified B-VAE. The results they get with GAN are poor, and the authors claim that GAN cannot work in their case. This is a very strong, unscientific statement, disproven by other papers on similar cases like Musella and Gandolfi arXiv:1805.00850 or Di Sipio et al arXiv 1903.02433.”

→ We agree with the sentiment. We significantly tone down the generality of our statement, compare to Musella and also test the DijetGAN on our ttbar data.

“The only statement they can make is that they did not manage themselves.”

→ We did that and included recent work in the ML literature that addresses the general question of how good GANs can learn densities.

“The statement (page 4 left) “Since the discriminator evaluates single events or small batches of events (with minibatch discrimination) in the training process, it is typically non-optimal to find differences in the densities of the training data and the generated data.” is plain wrong (if true GANs would never work, but they do).”

→ Statement is removed and the discussion of GANs is improved.

“Hence I strongly require that the GAN part of the paper be altogether removed. And then put B-VAE clearly in the title.”

→ Since the other reviewers were interested in a better comparison to bigger and more recent GAN architectures we have expanded our investigation to this end and hope that you are happy with the improved study and discussion.

General suggestion

“The tone of the paper is sometimes aggressive and should be toned down. “

→ We tried to use a less aggressive tone.

Specific suggestions

“I

[1-8] : not sure that 4 should be cited just there

[0-13] : not sure that 13 should be cited there”

→ We agree with 4. We understand the concerns with 13 in that context: image generation was not the primary objective but image generation was used to solve the primary objective: generating EFT models by training a GAN on holomorphic

functions represented as image data. For this reason we think it is appropriate to cite 13 here.

“Page 2 left column

“We find that only by using a density” : there is no way to prove this is true. Just say “We find that by using a density” “

→ Agreed! Adjusted.

““We introduce a degeneracy” : please spell it out”

→ Done!

“Page 2 Right column

b

Please list unambiguously all the variables to be learned (like you did for (a))

c

Same comment”

→ Done!

“Page 3 left column

Equation 4 MSE and DKL should be indicated two sentences earlier

Page 3 right column

“Large N is favorable” => what value of N is used ?”

→ The corresponding paragraph was completely changed. However, in the corresponding part of the manuscript we have specified the number of events we have used for constructing the density information buffer.

“Page 4 right column

“the standard VAE” (several times)=> “the VAE” (not sure “standard VAE” has any meaning)”

→ We have kept “standard” in front of some “VAE” occurrences to allow for a better distinction between the B-VAE and the staged VAE. We have looked at a version where “standard” was removed and found it to read strange. If you insist, we can of course provide you with a version not containing “standard VAE” to judge again.

“However, we must note that the corresponding widths” => where is this shown ???

“Fig 5 and 6 and 10 : indicate in the caption which parameter tuning is the best.”

→ Figs 5 and 6 changed to one Figure containing only the best model vs. ground truth distributions. Fig. 10 is not used in the context of finding optimal hyperparameters of the B-VAE but instead used as a) a simple application to experimental data and b) as an example for a small data scenario (20 000 events) which is discussed in the conclusions to pinpoint a weakness of the method.

“Page 5 left column

“increased diversity among events “ how do you see this ? is this not seen in Fig 7 introduced late ?”

→ Yes, it is seen in Fig. 7 of the old document. We have modified the whole discussion of the results and have made this point clearer.

“ 10^8 faster : please remove (also from the abstract). Rather only says sthg like :to be compared to up to 10-100s per event.”

→ Changed to “several orders of magnitude faster”

“Fig 7 : would the right column not be more clear with contours ??? Top line : change the color rule, there is no reason the MC to be blue because of less events.”

→ Changed the color code. We are not so sure about the contours and which are meant specifically. For the purpose of illustrating how the bins are filled up by the B-VAE in comparison to the MC generator we think the modified figure is suited well.

““for the first time”. Remove ! already done in <https://arxiv.org/pdf/1903.02433.pdf> “

→ Done!

“Page 5 right column

The whole paragraph “ An application of GAN.... Supersymmetry and Z” : sorry I just do not understand. Remove.”

→ Done!

“Conclusion

“for the first time” => remove!”

→ Done!

“The value and impact of the paper would be significantly increased if the code was released.”

→ Done!

“All figures: make sure the unit of the axis is always indicated”

→ Done!

“Bibliography :

Add Musella and Gandolfi arXiv:1805.00850 and Di Sipio et al arXiv 1903.02433”

→ Done! We also investigated how the DijetGAN in the Di Sipio et al. paper performs on our data.

List of changes:

We will not mention again all of what was reported in response to individual referee remarks but provide a brief overview.

- Abstract has been adjusted
- Several small changes to existing text in the introduction.
- Significant extension of the introduction incorporating remarks and other efforts on event generation with machine learning
- Provided more clarity on the toy model data and the composition / dimensionality of all investigated datasets
- Reformulation and much deeper presentation of the VAE methodology
- Significant changes to the Regular GAN presentation
- Extension of methodology to incorporate several GAN models
- Inclusion of traditional density estimation methods
- Explicit treatment of the latent code density estimation and the density information buffer in individual subsections
- An additional subsection on figures of merit
- Added sanity checks: a) provided baseline on event generation with Kernel Density Estimation, Gaussian Mixture Models and a Gaussian smearing of events (without any encoder/decoder) b) Checked the reconstruction of noise and compared to the reconstruction of test data events
- Added systematic study on the B-VAE hyperparameters: dimension of latent space, B , α and γ w.r.t. figures of merit
- Applied several density estimation methods on the encoded ground truth data (Kernel Density Estimation, Gaussian Mixture Models, Staged VAE, aggregated prior)
- Applied several GAN models on the $t\bar{t}$ data
- Major revision of applications and conclusion section

Reviewers' comments:

Reviewer #1 (Remarks to the Author):

I acknowledge that the authors of this manuscript have addressed the comments to the initial draft. By doing so, the text underwent a major revision, which unfortunately worsened the overall comprehensibility of its main messages.

In the following, the main messages are listed, commented or criticised as well as recommendations towards possible improvements in their presentation are given.

B-VAE:

The main scientific novelty of this work is the usage of a Density Information Buffer in the Variational Auto Encoder which is referred to as B-VAE in this work. The authors are able to motivate this approach and they put it into the context of existing methods citing the appropriate literature hereby. They provide instructions regarding its implementations and compare it to alternative density estimation techniques as requested. This meets the requirement of being a novelty and thus it should eventually justify a publication.

Unsuccessful attempts with GANs:

At the same time, the authors show, by ways of example, that their B-VAE is superior to existing generative methods, in particular to GANs. With respect to their initial submission, they have extended their investigations to more recent GAN-based techniques. In this sense, their conclusions are, in fact, founded on more solid evidence. Nonetheless, the applied strategy is not yet fully conclusive as the performance of any GAN is typically influenced strongly by the choice of the involved hyperparameters. For instance, the lambda parameter for the WGAN-GP in Line 398 requires application-specific tuning which was not done here. At least this is not mentioned here. Optimisation of such parameters for all elaborated GAN techniques seems to be beyond the scope of this work. Therefore, one of the following two alternatives is recommended:

1. Focussing on one of the state-of-the-art GAN techniques (e.g. WGAN-GP) and finding the optimal set of hyperparameters and network architectures for comparison to the B-VAE.
2. The authors do not provide non-trivial, original, i.e. not taken from existing literature, ideas on how to generally improve the GAN performance. Thus, one could limit the discussions on GANs in this publication to a brief note on the author's unsuccessful attempts to deploy them in the context of physics event generation.

Choice of figures of merit and performance assessment:

The figures of merit in equations 35 and 36 are barely motivated. They are not novel since they are computed from a logarithmic sum of already existing quantifiers. In my opinion, the construction of these figures of merits is poor due to the subsequent reasoning:

1. In the sum of equation 35, it is not clear what values the individual terms take and which one typically dominates the sum. What is the purpose of each term? Why is the simple chi², or the

Wasserstein distance, or the JSD alone not sufficient? Why is it reasonable to give all physics observables the same weight in this quantifier? Aren't some observables more important than others?

2. The figure of merit as defined by equation 36 is too sensitive to the fraction of holes. If, by chance, the fraction of holes is identical between MC and the NN, the chi2 and JSD could be arbitrarily high (e.g. due to completely mis-modelled correlations) but this quantifier would still be equal to 0 hereby incorrectly indicating good performance. By the way, what is the reason for not including the Wasserstein distance here? This should be explained in the text.

Concerning the binning, it is not motivated why 25 bins are chosen for the one-dimensional considerations but 1000x1000 (=1 million) bins for the two-dimensional investigations. 1 million bins for 1.2 million evaluation samples result in an average bin content of only 1.2. Consequently, the number of bins appears to be too high for a reliable statistical evaluation.

The NaN's that occur at multiple occasions in the evaluation are not acceptable and need to be removed or their origin must be explained and be interpreted properly.

PCA application for specifying the phase space:

I am still not convinced that the proposed PCA-based method is goal oriented. In practice, one does not want to generate events from one distinct point in the physical phase space, meaning events that are similar to one particular event (e.g. "event 62", cf. Line 995). Instead one is interested in generating events in which certain objects (such as an electron) fulfil some threshold criteria (e.g. having more than 20 GeV/c in transverse momentum and less than a given pseudorapidity). Such selections are equivalent to restricting the analysis to phase space volumes instead of phase space points. Hence what is ultimately desired is rather a mapping of latent space volumes to physical phase space volumes and not latent space points to physical phase space points. I would like the authors to comment on this.

Applications beyond:

The authors mention possible applications of their B-VAE method for MC integration or anomaly detection. However, they do not provide concrete examples nor any results of such. Hence, claiming to have "discovered the potential of the B-VAE to accelerate ..." (cf. Lines 1128-1130) is simply an exaggeration.

Concerning the anomaly detection, the authors should outline an actual implementation of this approach. Namely, the different Standard Model (SM) cross sections of interesting interaction processes in the proton-proton collisions span multiple orders of magnitude. In this sense, even a production of a Higgs boson pair (very rare) may appear as an anomaly with respect to the white SM background. Therefore: How big does the training dataset need to be in order to be sensitive to really new physics beyond the SM?

Secondly, if the anomaly detector is set up on simulated data, is MC-Data agreement an issue? How can one exclude that the anomaly detector is not sensitive to non-simulated detector effects? The authors should at least briefly discuss these points in the text.

Furthermore, I have some general comments related to the overall structure and presentation:

Usage of different datasets:

My initial comment remains: “(...) the individual studies on each dataset are linked to different network architecture- and training tunings aggravating the reproducibility of the results. (...)”

In fact, due to the introduction and investigation of more GAN-techniques and the figures of merit, the mix of performance assessments, different modelling techniques plus numerous network architectures as a function of the dataset inhibits the comprehensibility of the presentation.

I strongly recommend the usage of one dataset, ideally ttbar or the CMS open dataset, on which the different generative methods with a consistent, common evaluation are studied.

Any pre- and post-processing of the datasets should be defined only once and not for every method individually.

Usage of different network architectures:

The comparison between the network architectures would benefit from choosing similar network structures, e.g. with the same number of hidden layers, with the same number of nodes, same activation functions, etc. Deviations from the nominal architecture, such as method-specific normalisations (e.g. why is layer normalisation only used for MMD-GANs, cf. Line 425?), should be clearly motivated.

By this means, the architecture specifications would be disentangled from the training metric which facilitates the interpretation of the results.

It would also be helpful to show the evolution of the different loss curves during the training.

Review of generative modelling techniques:

The explanations of the various generative modelling techniques is too detailed and should be shortened by properly citing the existing literature. For instance, the challenge of mode collapsing for “regular” GANs has been described in other works already.

Moreover, the phrase “(...) we find (...)” in Line 454 and others reads as if the authors of this work have originally performed these general derivations themselves. I recommend a more neutral expression such as “one finds that (...)” if the equations are quoted from elsewhere. This comment holds for all subsequent passages.

Sanity checks:

I personally have trouble putting the sanity checks in the general context of this work. They are not outlined nor announced anywhere before but appear rather ad hoc. The authors should verify if the sanity checks are strictly necessary in this work and if so elaborate more on their importance and the

consequences for the subsequent results.

Figures:

The figures need to be improved. Axis labels and legends need to be clearly visible which is not the case when printing the manuscript on A4 paper.

The upper graphics in Figure 6 do not convey any message as they are fully black.

In summary, the submitted manuscript does not justify a publication at its current form. While the B-VAE is an interesting and novel concept introduced in this work, some claims as well as the used figures of merit require careful revision.

I find the overall structure of this work rather confusing. In particular, the authors should consider the usage of one dataset, to homogenise network architectures, training algorithms and the performance assessments.

Some statements are not fully convincing and require more justification. More detailed however informal comments to individual passages of the text are attached.

Appendix:

- This review incl. informal comments to individual passages of the text

Reviewer #2 (Remarks to the Author):

Please see my original review regarding the novelty and merit of the paper.

The revised manuscript is greatly improved, both in terms of the experimental results which support the authors' claims, and in terms of the writing of the paper. Highlights of the improvements in the revision include:

-Added comparisons to a range of more recently proposed GAN variants. The overall claims have been tempered but are now much better supported by the experimental results. (Roughly, the authors have now tried quite hard to address the problem using a broad range of GAN variants, and despite these efforts have still found that GAN-style methods did not succeed to the degree of the novel VAE variant that they have proposed.)

-An improved discussion of GAN and VAE methods, including a better explanation of the strengths, weaknesses, and mathematical derivation of the proposed method.

-A comparison to traditional density estimation methods (kernel density estimation, mixture of Gaussians).

-A more rigorous quantitative evaluation which addresses the potential of the proposed method to

overfit, including the introduction of, and evaluation via, novel metrics which take overfitting into account ("figures of merit").

-Numerous minor textual issues have been corrected.

I appreciate the authors' diligent efforts. I now support the publication of the manuscript. I just have a few very minor suggestions for the final version:

-Regarding Reviewer 1's point, "the authors fail to provide a generic recipe of their PCA-based method rather than demonstrating it on one example only": While the authors' method here is clear based on their case study, it would be informative --and easy to do-- for the authors to further provide a pseudo-code figure containing the steps that they performed in their PCA-based case study, as a general recipe for how practitioners should apply this method in other contexts and datasets.

-Page 2 (line 88), "a approximate agreement"  "an approximate agreement"

-Page 7 (line 589), "We slightly modify the loss function": Explain why this is necessary in this context.

-Page 7, the paragraph below Equation 29 (line 604): Explicitly define μ^i , $\sigma^{2,i}$.

-In Figure 8, I do not see any yellow points for DijetGAN.

-Some of the experiments compare numerical properties of the generated samples to those of the real data, such as the fraction of "holes" or the frequencies of certain events. A principled way to measure the agreement between simulated data under a Bayesian model, and the real data, according to a chosen statistic of interest (e.g. the fraction of "holes"), is to use posterior predictive checks:

Gelman, A., Meng, X. L., & Stern, H. (1996). Posterior predictive assessment of model fitness via realized discrepancies. *Statistica sinica*, 733-760.

The authors may consider using posterior predictive checks as a tool to quantify the ability of the models to reproduce the relevant and desired properties of the real data. This would augment the results which are already in the paper. This is simply a suggestion, and I do not insist on it for the paper to be accepted.

Reviewer #3 (Remarks to the Author):

The manuscript is much improved compared to the first submission, and the authors have followed the recommendations of the referees (at least mine). Also the authors made available the data and the code as was requested. I can now recommend it for publication as is.

Nature Communications, response to referees for NCOMMS-19-17549B

Dear referees,

we want to thank all of you for your patience and the time you took to review our work. We want to apologize for the delay that occurred due to the pandemic. We hope that you are healthy and happy with the adjustments we have made.

Reviewers' comments:

Reviewer #1 (Remarks to the Author):

I acknowledge that the authors of this manuscript have addressed the comments to the initial draft. By doing so, the text underwent a major revision, which unfortunately worsened the overall comprehensibility of its main messages.

In the following, the main messages are listed, commented or criticised as well as recommendations towards possible improvements in their presentation are given.

>>> We have put more effort into improving the overall comprehensibility while conserving all expressed interests by the referees.

B-VAE:

The main scientific novelty of this work is the usage of a Density Information Buffer in the Variational Auto Encoder which is referred to as B-VAE in this work. The authors are able to motivate this approach and they put it into the context of existing methods citing the appropriate literature hereby. They provide instructions regarding its implementations and compare it to alternative density estimation techniques as requested. This meets the requirement of being a novelty and thus it should eventually justify a publication.

>>> Thank you.

Unsuccessful attempts with GANs:

At the same time, the authors show, by ways of example, that their B-VAE is superior to existing generative methods, in particular to GANs. With respect to their initial submission, they have extended their investigations to more recent GAN-based techniques. In this sense, their conclusions are, in fact, founded on more solid evidence. Nonetheless, the applied strategy is not yet fully conclusive as the performance of any GAN is typically influenced strongly by the choice of the involved hyperparameters. For instance, the lambda parameter for the WGAN-GP in Line 398 requires application-specific tuning which was not done here. At least this is not mentioned here. Optimisation of such parameters for all elaborated GAN techniques seems to be beyond the scope of this work. Therefore, one of the following two alternatives is recommended:

1. Focussing on one of the state-of-the-art GAN techniques (e.g. WGAN-GP) and finding the optimal set of hyperparameters and network architectures for comparison to the B-VAE.

2. The authors do not provide non-trivial, original, i.e. not taken from existing literature, ideas on how to generally improve the GAN performance. Thus, one could limit the discussions on GANs in this publication to a brief note on the author's unsuccessful attempts to deploy them in the context of physics event generation.

>>> Essentially, we have taken path 2. We significantly reduced the presence / importance / generality of the GAN results in our paper. We no longer mention the unsuccessful attempts with GANs in the abstract and have cut away all the details on the GAN theory for individual architectures and training procedures / architectural details. In several places we have made clear that we tried GANs only with the recommended hyperparameters and did no application-specific tuning. We did, however, spend a similar amount of CPU time (compared to VAE architectures) in testing the GAN architectures.

Choice of figures of merit and performance assessment:

The figures of merit in equations 35 and 36 are barely motivated. They are not novel since they are computed from a logarithmic sum of already existing quantifiers. In my opinion, the construction of these figures of merits is poor due to the subsequent reasoning:

1. In the sum of equation 35, it is not clear what values the individual terms take and which one typically dominates the sum. What is the purpose of each term? Why is the simple χ^2 , or the Wasserstein distance, or the JSD alone not sufficient? Why is it reasonable to give all physics observables the same weight in this quantifier? Aren't some observables more important than others?

>>> The referee is correct: the χ^2 , the Wasserstein distance and the JSD all show more or less the same behavior. We have therefore restricted our figure of merit to χ^2 only. For simplicity we have given the same weight to all observables. This can be changed for other purposes.

2. The figure of merit as defined by equation 36 is too sensitive to the fraction of holes. If, by chance, the fraction of holes is identical between MC and the NN, the χ^2 and JSD could be arbitrarily high (e.g. due to completely mis-modelled correlations) but this quantifier would still be equal to 0 hereby incorrectly indicating good performance. By the way, what is the reason for not including the Wasserstein distance here? This should be explained in the text.

>>> We have now separated the fraction of holes from the 2D figure of merit by introducing a third figure of merit δ_{OF} that is only concerned with the fraction of holes. For the 2D case we now apply a more rigorous procedure by calculating the χ^2 for all 2D histograms between observables. Similarly, we remove mentioning of the JSD and Wasserstein distance (which yield the same conclusions).

Concerning the binning, it is not motivated why 25 bins are chosen for the one-dimensional considerations but 1000x1000 (=1 million) bins for the two-dimensional investigations. 1 million bins for 1.2 million evaluation samples result in an average bin content of only 1.2. Consequently, the number of bins appears to be too high for a reliable statistical evaluation.

>>> We compute δ_{1D} and δ_{2D} now for 25 bins and 25^2 bins respectively. For δ_{OF} we integrate the absolute difference between the fraction of holes in the MC data and the NN data over the $\sqrt{N_{bins}}$ up until 1000×1000 bins.

The NaN's that occur at multiple occasions in the evaluation are not acceptable and need to be removed or their origin must be explained and be interpreted properly.

>>> NaNs no longer occur.

PCA application for specifying the phase space:

I am still not convinced that the proposed PCA-based method is goal oriented. In practice, one does not want to generate events from one distinct point in the physical phase space, meaning events that are similar to one particular event (e.g. "event 62", cf. Line 995). Instead one is interested in generating events in which certain objects (such as an electron) fulfil some threshold criteria (e.g. having more than 20 GeV/c in a transverse momentum and less than a given pseudorapidity). Such selections are equivalent to restricting the analysis to phase space volumes instead of phase space points. Hence what is ultimately desired is rather a mapping of latent space volumes to physical phase space volumes and not latent space points to physical phase space points. I would like the authors to comment on this.

>>> We agree with the referee and have clarified in the text.

Applications beyond:

The authors mention possible applications of their B-VAE method for MC integration or anomaly detection. However, they do not provide concrete examples nor any results of such. Hence, claiming to have "discovered the potential of the B-VAE to accelerate ..." (cf. Lines 1128-1130) is simply an exaggeration.

>>> We have reformulated the exaggeration: we now only propose a more detailed investigation in that context and claim to provide hints for why this should be fruitful.

Concerning the anomaly detection, the authors should outline an actual implementation of this approach. Namely, the different Standard Model (SM) cross sections of interesting interaction processes in the proton-proton collisions span multiple orders of magnitude. In this sense, even a production of a Higgs boson pair (very rare) may appear as an anomaly with respect to the white SM background. Therefore: How big does the training dataset need to be in order to be sensitive to really new physics beyond the SM?

Secondly, if the anomaly detector is set up on simulated data, is MC-Data agreement an issue? How can one exclude that the anomaly detector is not sensitive to non-simulated detector effects? The authors should at least briefly discuss these points in the text.

>>> We have added a brief paragraph on the hint we find for why this could be a useful anomaly detector by interpreting Fig. 1 (i.e. unseen events are reconstructed very well and noise is reconstructed badly, learning the reconstruction is enhanced with small beta). Details for the BSM physics case of anomaly detection are presented in the Les Houches new physics working group report of 2019. In the submitted paper we merely

want to suggest the investigation of the B-VAE for this purpose. Testing the B-VAE (compared to other methods) for anomaly detection is the purpose of upcoming work.

Furthermore, I have some general comments related to the overall structure and presentation:

Usage of different datasets:

My initial comment remains: “(...) the individual studies on each dataset are linked to different network architecture- and training tunings aggravating the reproducibility of the results. (...)”

In fact, due to the introduction and investigation of more GAN-techniques and the figures of merit, the mix of performance assessments, different modelling techniques plus numerous network architectures as a function of the dataset inhibits the comprehensibility of the presentation.

I strongly recommend the usage of one dataset, ideally ttbar or the CMS open dataset, on which the different generative methods with a consistent, common evaluation are studied.

>>> We had a hard time to decide on how to react to this since the other two referees seemed to appreciate the broader study involving several datasets and models. However we also have seen how the presentation may have been a mix of too many elements in which the key messages may get lost. Therefore we have tried hard to find a good compromise:

- Most importantly, we want to show this for different (increasingly) higher dim problems. GAN might still work for 6d (as shown in dijet GAN paper)
- in a first step we have removed all of the GAN theory referring only to the papers and the recommended hyperparameters therein.
- in several places we put more emphasis on explaining the value of the finding such that the B-VAE and its investigation on the ttbar data is really the main part of the paper.
- we have tried to improve the orientation of the reader in the paper by adding a brief summary of what the subsequent chapters will be about at the end of the introduction.
- we drastically reduced the amount of network architectures by excluding all GAN network architectures and training procedures. In the whole paper, there are now only two network architectures, both for the B-VAE, and the only difference is the number of layers (4 for two-body & Z decay, 6 for ttbar). This choice was made a priori because we assumed that modelling tt-bar in 26d will require more neural capacity / network complexity than the two-body & Z decay. This difference however is relatively insignificant and can be found in the code in the GitHub repository as well, so we can also leave out this detail but we left it in for now.
- to further improve comprehensibility, we removed some redundant / irrelevant passages and refined some formulations.
- Two tables (those with the ranking in the 2d f.o.m) became redundant after adjusting the figures of merit and were therefore removed.

Any pre- and post-processing of the datasets should be defined only once and not for every method individually.

>>> done.

Usage of different network architectures:

The comparison between the network architectures would benefit from choosing similar network structures, e.g. with the same number of hidden layers, with the same number of nodes, same activation functions, etc. Deviations from the nominal architecture, such as method-specific normalisations (e.g. why is layer normalisation only used for MMD-GANs, cf. Line 425?), should be clearly motivated.

By this means, the architecture specifications would be disentangled from the training metric which facilitates the interpretation of the results.

>>> Although we were generally following the outlines for the architectures in the corresponding papers, we have removed the details in this regard for the sake of comprehensibility.

It would also be helpful to show the evolution of the different loss curves during the training.

>>> Added

Review of generative modelling techniques:

The explanations of the various generative modelling techniques is too detailed and should be shortened by properly citing the existing literature. For instance, the challenge of mode collapsing for “regular” GANs has been described in other works already.

Moreover, the phrase“(...) we find (...)” in Line 454 and others reads as if the authors of this work have originally performed these general derivations themselves. I recommend a more neutral expression such as “one finds that (...)” if the equations are quoted from elsewhere. This comment holds for all subsequent passages.

>>> Agreed and changed.

Sanity checks:

I personally have trouble putting the sanity checks in the general context of this work. They are not outlined nor announced anywhere before but appear rather ad hoc. The authors should verify if the sanity checks are strictly necessary in this work and if so elaborate more on their importance and the consequences for the subsequent results.

>>> We announce them in the introduction now. They are not strictly necessary but a) interesting and b) answered concerns raised by other referees.

Figures:

The figures need to be improved. Axis labels and legends need to be clearly visible which is not the case when printing the manuscript on A4 paper. The upper graphics in Figure 6 do not convey any message as they are fully black.

>>> These figures are improved. We agree with the referee. That is why we had blue points on white background resembling the data in the upper graphics in figure 6 in the previous version but another referee commented that there would be no reason to choose different coloring: as a synthesis we now have the same custom colormap for all plots in that figure.

In summary, the submitted manuscript does not justify a publication at its current form. While the B-VAE is an interesting and novel concept introduced in this work, some claims as well as the used figures of merit require careful revision.

I find the overall structure of this work rather confusing. In particular, the authors should consider the usage of one dataset, to homogenise network architectures, training algorithms and the performance assessments.

Some statements are not fully convincing and require more justification. More detailed however informal comments to individual passages of the text are attached.

>>> We thank the referee for the careful review and hope that they are happy with the improved version.

Appendix:

- This review incl. informal comments to individual passages of the text

Reviewer #2 (Remarks to the Author):

Please see my original review regarding the novelty and merit of the paper.

The revised manuscript is greatly improved, both in terms of the experimental results which support the authors' claims, and in terms of the writing of the paper. Highlights of the improvements in the revision include:

-Added comparisons to a range of more recently proposed GAN variants. The overall claims have been tempered but are now much better supported by the experimental results. (Roughly, the authors have now tried quite hard to address the problem using a broad range of GAN variants, and despite these efforts have still found that GAN-style methods did not succeed to the degree of the novel VAE variant that they have proposed.)

-An improved discussion of GAN and VAE methods, including a better explanation of the strengths, weaknesses, and mathematical derivation of the proposed method.

-A comparison to traditional density estimation methods (kernel density estimation, mixture of Gaussians).

-A more rigorous quantitative evaluation which addresses the potential of the proposed method to overfit, including the introduction of, and evaluation via, novel metrics which take overfitting into account ("figures of merit").

-Numerous minor textual issues have been corrected.

I appreciate the authors' diligent efforts. I now support the publication of the manuscript. I just have a few very minor suggestions for the final version:

>>> We thank the referee for their good summary of the changes and the support of the publication.

-Regarding Reviewer 1's point, "the authors fail to provide a generic recipe of their PCA-based method rather than demonstrating it on one example only": While the authors' method here is clear based on their case study, it would be informative --and easy to do-- for the authors to further provide a pseudo-code figure containing the steps that they performed in their PCA-based case study, as a general recipe for how practitioners should apply this method in other contexts and datasets.

>>> We agree that the recipe is easy to provide and sort of obvious. In any case, we hope to have settled the concerns with the new response to referee 1 in this regard.

-Page 2 (line 88), "a approximate agreement"  "an approximate agreement"

>>> Changed.

-Page 7 (line 589), "We slightly modify the loss function": Explain why this is necessary in this context.

>>> done.

-Page 7, the paragraph below Equation 29 (line 604): Explicitly define μ^i , $\sigma^{2,i}$.

>>> done.

-In Figure 8, I do not see any yellow points for DijetGAN.

>>> Plot no longer exists.

-Some of the experiments compare numerical properties of the generated samples to those of the real data, such as the fraction of "holes" or the frequencies of certain events. A principled way to measure the agreement between simulated data under a Bayesian model, and the real data, according to a chosen statistic of interest (e.g. the fraction of "holes"), is to use posterior predictive checks:

Gelman, A., Meng, X. L., & Stern, H. (1996). Posterior predictive assessment of model fitness via realized discrepancies. *Statistica sinica*, 733-760.

The authors may consider using posterior predictive checks as a tool to quantify the ability of the models to reproduce the relevant and desired properties of the real data. This would augment the results which are already in the paper. This is simply a suggestion, and I do not insist on it for the paper to be accepted.

>>> Thank you for this suggestion! We will keep it in mind as a possible augmentation for the future. We hope that the modifications we performed to respond to the concerns of referee 1 also improve the quality of the paper for you.

Reviewer #3 (Remarks to the Author):

The manuscript is much improved compared to the first submission, and the authors have followed the recommendations of the referees (at least mine). Also the authors made available the data and the code as was requested. I can now recommend it for publication as is.

>>> Thank you very much! We hope that you are still happy with the manuscript.

REVIEWER COMMENTS

Reviewer #1 (Remarks to the Author):

Dear editor,
dear authors,

I thank the authors for properly addressing my general comments to the previous draft of this paper. I acknowledge that not all of my suggestions were incorporated though in order to avoid potential conflicts with the reviews by the other two referees.

Nonetheless, I am pleased to see that the major reasons for which the previous draft was rejected, e.g. the “NaNs” in the performance tables, were dealt with. By extensively shortening the elaborations on GANs, the comprehensibility of this work has been boosted overall. Altogether, this draft is now converging towards a publishable article.

However, there are still some issues with the text for which another (minor) revision is unavoidable. My main concerns with this version are described in the following:

1.) The chi2 to quantify the agreement between histograms is a widely used measure. Contrary to the authors' claim of novelty, the chi2 is also used in other works (e.g. Ref. 20) to quantify the agreement between “real” and “generated” distributions. More importantly, the usage of a logarithm is cumbersome and not straightforward to grasp. What happens for instance in the case of ideal agreement between histograms, i.e. when $\chi^2 = 0$? Then, the logarithm converges to negative infinity. Negative infinity would also be reached if ideal agreement is achieved for only one of the eleven observables in the ttbar dataset while the other ten could have arbitrary levels of disagreements. On the other hand, the natural chi2, i.e. without logarithm, has a lower bound of 0 and is easier to understand. As the authors state, the neglect of the logarithm leads to similar results. So there is no added benefit of making the quantifier more complicated. In addition, what is the relative importance between the three indicators? By considering one hyper-parameter setting to be the best, the authors implicitly deploy a weighting scheme. This should be quantified or at least elaborated further,

Last but not least, one should outline how the ideal performance indicator values of 11, 8 and 0.5 are obtained. In summary, the performance quantifications are not presented in a quite convincing manner.

2.) Some passages need further specification. One example: In lines (L.) 367-379 it is not evident for which datasets the preprocessing is applied. See more examples in the detailed comments at the bottom.

3) MC datasets themselves are not a method - and should not be part of the methodology. It should be considered to move the description of the datasets into an own section.

4) The presentation of the results appears a bit confusing due to the random sequence of used datasets. The argumentation line that stuck after reading this section is the following: Sanity checks with ttbar and B-VAE, preliminary studies with toy model and $Z \rightarrow ll$ (with, in my opinion, no added

benefit of the toy model) using different generative methods, evaluation and tuning of B-VAE on ttbar, demonstration that GANs are not suitable shown on $Z \rightarrow ll$. What would be better is: GANs unsuitable on $Z \rightarrow ll$, sanity checks of B-VAE with ttbar, tuning and performance assessment with ttbar and B-VAE.

5) The language in some passages needs to be improved for publication in a renowned journal. In particular the nomenclature is inconsistent (e.g. fake events = generated events = artificial events, or particle physics events = MC events = real events = MC event data = physical events = ..., ...), the usage of present, past and present perfect tenses need to be revised, and a non-negligible amount of particle physics “slang” can be found throughout the text (e.g. “event smearing”, “Standard Model cocktail”, “when plotting”).

6) The quality of the figures has been improved as requested. However, every figure should still have proper axis labels (e.g. Fig. 6c and Fig. 8).

Considering all suggestions, the overall structure and content of this work should not change too much anymore.

Therefore, I invite the authors (addressed directly in the following) this time to also respond to my detailed comments / suggestions or questions that are stated below. Please excuse potential overlap therein with the general concerns 1)-6) above.

Detailed comments / suggestions / questions:

L. 17 - 18: As the meaning of “physical events” may be ambiguous in the broad physics community, one should define the meaning of physical events to be high energy particle collisions right at the beginning.

L. 51: Technically, this statement is not quite accurate. The model does not learn $p(x)$ but rather a transformation $z \rightarrow x$ such that the distribution of x follows $p(x)$.

L. 61 - 65: This is not a sentence. A verb is missing.

L. 76: Only the ttbar dataset in your study has higher dimensionality than 8. Hence, the $Z \rightarrow ll$ (2x8 dimensions) and the toy model will not help investigating whether “generative models are able to reliably model processes with a larger number of objects.” So why is the dimensionality argument emphasised here?

L. 114-117 (and later again): The counting of physics objects is not clear. ttbar+1lepton has 1 lepton + 4 jets + MET (MET is typically considered an object, namely as neutrino(s)) = 6 while ttbar+2leptons has 2 leptons + 2 jets + MET = 5. The total count agrees with what is written here. But, in L. 155 or L. 225 - 226 you count 4 jets + 1 (2) leptons = 5 (6) objects. Even later, you speak of maximum of 4 jets (which is correct). Please be consistent to avoid such unnecessary confusions.

L. 120: A “.” Is missing.

L. 141: Missing “,”.

L. 189: Top production = slang. Better: top quark production.

L. 201: What is the benefit of keeping electrons and muons separate in the $Z \rightarrow ll$ decay? For the ttbar dataset you just characterise them as leptons. Please elaborate. Furthermore, as you do not run any detector simulation for this dataset, why do you not consider all possible Z-boson decay modes, e.g. taus and quarks as well?

L. 233 and L. 235: The wording is confusing. “the latter” (L. 233) and “the B-VAE” (L. 235) are the same thing, right? If so, I would drop the “For the B-VAE” part.

L. 239 - 240: Please indicate the respective versions of the used software packages. This would also be helpful for the README in the public code repository on github.

L. 249 - 250: I am not sure what the specification “learned” event generators means? I think the term “learned” can be dropped.

L. 250: It is not clear what “these” refers to.

L. 251: A citation for the Jensen-Shannon GAN is missing.

L. 259: This should be the start of a new sentence.

L. 264 - L. 265: Why is it relevant to state that the training time of both approaches was comparable?

L. 320 - 322: You state that it should be sufficient to sample epsilon once. But are you also doing that or why is this explicitly mentioned here? Please be more specific.

L. 367 - L. 379: It is not clear from reading the text if these pre-processing procedures are applied on all datasets or not. If not, for which? Furthermore, where do you describe the architectures for the toy model?

L. 389: What does it mean if a distribution is not correct? What you rather intend to claim is that the generated distribution does not match the reference.

L. 420 - 421: “Evenly” or “uniformly” distributed? I think you mean uniformly.

L. 457: Is it important to state that the values are “comma-separated” in the file?

L. 495 - 498: I strongly disagree with the claim that previous works on generative modelling in physics focussed solely on simply comparing histograms. Correlations between reconstructed quantities are also investigated elsewhere. Even in the cited references, Ref. 20 quotes χ^2 for the agreement between observable distributions and Ref. 21 explicitly computes and compares pair-wise correlation coefficients between “real” and “generated” events.

L. 511: Please define what “leading” means.

Equation 20: Indexes i, j need to be defined as observable indexes.

L. 519: What do you mean by divergences? If you mean alternative definitions, state what those are.

L. 523 - 525: Ref. 45 does not suggest the usage of a logarithm. $\chi^2 \rightarrow 0$ in one observable distribution (=supreme accuracy) would lead to $-\infty$. Why do you insist on using it here, especially since the approach without the log does not alter the result?

L. 555: How are the ideal generator performance indicators obtained?

L. 574: Is this a general statement or one that applies for the investigated dataset only? If it is the latter, please add “in this study” or write “achieved by using the DijetGAN approach” or similarly.

L. 581 - 584: This is a repetition from L. 366. Please remove it.

L. 604 - 607: Again, this is a repetition because the pre-processing was described before.

L. 611: “smearing of events” is a slang which people outside the particle physics analysis community will likely have trouble understanding.

L. 617 - 618: Why are the smeared events close to the true events if all kinematic quantities are smeared by an additional 10% resolution? This is a non-trivial statement.

L. 618 - 620: If this procedure does not respect correlations, why does it not perform poorly on Δ^2 which itself is essentially a correlation measure?

L. 640: As there are no real insights presented and the toy model’s training & architecture are not elaborated before, one should consider removing it from this paper.

L. 670: “Plotting” = slang.

L. 715-716: Claiming that best performances in Δ_{2D} together with poor performances in

delta_1D provide small advantages imply a hierarchy among the performance indicators (delta_2D is more important than delta_1D). However, such a hierarchy is not discussed - instead it is said somewhere that they can be weighted arbitrarily.

L. 724: Why do you consider this choice to be the best? You implicitly make use of a weighting of performance indicators hereby which should be quantified somehow.

L. 748: After "Finally" one expects a paragraph/section to end. Yet, the presentation of the results continues afterwards. I would suggest to move this statement on the computing advantage to the end or even in an own subsection.

L. 769-771: This is a repetition from L. 743-745.

L. 846-847: What do you consider to be the difference between a good generator and a good density sampler? Why is it not sufficient to only write "good generator of collision data" (or similar)?

L. 912: "Standard Model cocktail" = slang.

L. 918 - 934: Please elaborate what the (potential) advantage of using real experimental data for this study is? I.e. what is the added benefit of using CMS open data to your work? A real benefit would be if you compared real CMS data to simulated CMS data to B-VAE generated data.

L. 938 - 939: "developed" is a bit too strong as you are mostly deploying standard chi2 measures to compare the histograms. My understanding is that only delta_OF can be considered a non-standard quantity.

L. 986 - 991: I criticised this statement already before. To me this reads like an inherent contradiction. You essentially explain: "The distribution is not flat due to the low statistics. But if the statistics are increased - it would still not be flat." Furthermore, why would the distribution in phi not be flat for N_events → infinity? Is this a feature of the underlying CMS detector from Fig. 10? Fig. 7: What does "window size" mean? Do you mean "training iterations"?

Dear editor,
dear authors,

I thank the authors for properly addressing my general comments to the previous draft of this paper. I acknowledge that not all of my suggestions were incorporated though in order to avoid potential conflicts with the reviews by the other two referees.

Nonetheless, I am pleased to see that the major reasons for which the previous draft was rejected, e.g. the “NaNs” in the performance tables, were dealt with. By extensively shortening the elaborations on GANs, the comprehensibility of this work has been boosted overall.

Altogether, this draft is now converging towards a publishable article.

>> Thank you for the acknowledgement and your efforts! We are happy that the intended improvement was perceived as such.

However, there are still some issues with the text for which another (minor) revision is unavoidable. My main concerns with this version are described in the following:

1.) The chi2 to quantify the agreement between histograms is a widely used measure. Contrary to the authors' claim of novelty, the chi2 is also used in other works (e.g. Ref. 20) to quantify the agreement between “real” and “generated” distributions. More importantly, the usage of a logarithm is cumbersome and not straightforward to grasp. What happens for instance in the case of ideal agreement between histograms, i.e. when $\chi^2 = 0$? Then, the logarithm converges to negative infinity. Negative infinity would also be reached if ideal agreement is achieved for only one of the eleven observables in the ttbar dataset while the other ten could have arbitrary levels of disagreements. On the other hand, the natural chi2, i.e. without logarithm, has a lower bound of 0 and is easier to understand. As the authors state, the neglect of the logarithm leads to similar results. So there is no added benefit of making the quantifier more complicated. In addition, what is the relative importance between the three indicators? By considering one hyper-parameter setting to be the best, the authors implicitly deploy a weighting scheme. This should be quantified or at least elaborated further. Last but not least, one should outline how the ideal performance indicator values of 11, 8 and 0.5 are obtained. In summary, the performance quantifications are not presented in a quite convincing manner.

>> We have removed the log and defined a ranking mechanism. We also explained how we obtain the ideal performance indicators.

2.) Some passages need further specification. One example: In lines (L.) 367-379 it is not evident for which datasets the preprocessing is applied. See more examples in the detailed comments at the bottom.

>> This is now clarified. See answers to comments at the bottom.

3) MC datasets themselves are not a method - and should not be part of the methodology. It should be considered to move the description of the datasets into an own section.

>> We agree and have moved the description of the datasets into an own section.

4) The presentation of the results appears a bit confusing due to the random sequence of used datasets. The argumentation line that stuck after reading this section is the following: Sanity checks with ttbar and B-VAE, preliminary studies with toy model and $Z \rightarrow ll$ (with, in my opinion, no added

benefit of the toy model) using different generative methods, evaluation and tuning of B-VAE on ttbar, demonstration that GANs are not suitable shown on $Z \rightarrow ll$. What would be better is: GANs unsuitable on $Z \rightarrow ll$, sanity checks of B-VAE with ttbar, tuning and performance assessment with ttbar and B-VAE.

>> We changed the ordering and now perform the sanity checks at the end. We still wanted to keep the toy model because it shows that GANs do work to some degree and we were able to reproduce and solve the issue with phi that was reported in other papers. We thought that the sanity checks should follow after the ttbar results because we do the check for the identity function with the best B-VAE model.

5) The language in some passages needs to be improved for publication in a renowned journal. In particular the nomenclature is inconsistent (e.g. fake events = generated events = artificial events, or particle physics events = MC events = real events = MC event data = physical events = ..., ...), the usage of present, past and present perfect tenses need to be revised, and a non-negligible amount of particle physics “slang” can be found throughout the text (e.g. “event smearing”, “Standard Model cocktail”, “when plotting”).

>> We went through all the slang that was specified and revised the language very generally. We took care of a consistent nomenclature and tenses. Lastly we sent the manuscript to a native speaker (Dr. Clara Nellist) for a final check.

6) The quality of the figures has been improved as requested. However, every figure should still have proper axis labels (e.g. Fig. 6c and Fig. 8).

>> done

Considering all suggestions, the overall structure and content of this work should not change too much anymore.

Therefore, I invite the authors (addressed directly in the following) this time to also respond to my detailed comments / suggestions or questions that are stated below. Please excuse potential overlap therein with the general concerns 1)-6) above.

Detailed comments / suggestions / questions: ^[1]_{SEP}

L. 17 - 18: As the meaning of “physical events” may be ambiguous in the broad physics community, one should define the meaning of physical events to be high energy particle collisions right at the beginning.

>> Done.

L. 51: Technically, this statement is not quite accurate. The model does not learn $p(x)$ but rather a transformation $z \rightarrow x$ such that the distribution of x follows $p(x)$.

>> Done.

L. 61 - 65: This is not a sentence. A verb is missing.

>> The verb was ‘manages’. Sentence is reformulated.

L. 76: Only the ttbar dataset in your study has higher dimensionality than 8. Hence, the $Z \rightarrow ll$ (2x8 dimensions) and the toy model will not help investigating whether “generative models are able to reliably model processes with a larger number of objects.” So why is the dimensionality argument emphasised here?

>> 1. The literature thus far had shown issues (explained in the lines following L76) even for such “low dimensional” cases, in particular with phi. With the toy model we show that we can solve this issue.

2. The $Z \rightarrow ll$ indeed can be viewed as 2×8 dimensions in the ideal case. However, input and output of the models for $Z \rightarrow ll$ is a 16d vector. As input we do have cleanly separated vectors of the form $(x_1, \dots, x_8, 0, \dots, 0)$ and $(0, \dots, 0, x_9, \dots, x_{16})$ but the model itself has to learn that there are two 8d disjunct submanifolds in that 16d space and produce output accordingly. Without a buffer, one obtains many four lepton events.

L. 114-117 (and later again): The counting of physics objects is not clear. $t\bar{t} + 1\text{lepton}$ has 1 lepton + 4 jets + MET (MET is typically considered an object, namely as neutrino(s)) = 6 while $t\bar{t} + 2\text{leptons}$ has 2 leptons + 2 jets + MET = 5. The total count agrees with what is written here. But, in L. 155 or L. 225 - 226 you count 4 jets + 1 (2) leptons = 5 (6) objects. Even later, you speak of maximum of 4 jets (which is correct). Please be consistent to avoid such unnecessary confusions.

>> The data is slightly different: we always have four jets but only sometimes have one or two leptons. This is due to the detector simulation and taking only the four leading jets + at least one and at most two leptons.

L. 120: A “.” Is missing.

>> Fixed

L. 141: Missing “,”.

>> Fixed

L. 189: Top production = slang. Better: top quark production.

>> Agreed, fixed

L. 201: What is the benefit of keeping electrons and muons separate in the $Z \rightarrow ll$ decay? For the $t\bar{t}$ dataset you just characterise them as leptons. Please elaborate. Furthermore, as you do not run any detector simulation for this dataset, why do you not consider all possible Z-boson decay modes, e.g. taus and quarks as well?

>> The intention here was simply to tackle a problem that is more challenging than the toy model by adding the difficulty of dimensionality and learning two separable distributions at once. The setup would also allow decays like $Z \rightarrow e\mu\mu$ (i.e. 4 leptons), which actually happens without the buffering. We added a sentence to the paper.

L. 233 and L. 235: The wording is confusing. “the latter” (L. 233) and “the B-VAE” (L. 235) are the same thing, right? If so, I would drop the “For the B-VAE” part.

>> Yes and agreed, fixed.

L. 239 - 240: Please indicate the respective versions of the used software packages. This would also be helpful for the README in the public code repository on github.

>> Done.

L. 249 - 250: I am not sure what the specification “learned” event generators means? I think the term “learned” can be dropped.

>> Agreed, dropped “learned”.

L. 250: It is not clear what “these” refers to.

>> Clarified in text.

L. 251: A citation for the Jensen-Shannon GAN is missing.

>> Fixed

L. 259: This should be the start of a new sentence.

>> Agreed and fixed.

L. 264 - L. 265: Why is it relevant to state that the training time of both approaches was comparable?

>> We thought it is interesting. Due to the difference in the results, we felt it is noteworthy that the computational time investments were similar to avoid the impression that we just didn’t invest a fair amount of time. In the present version we have however still removed the sentence because after stating that the GANs just serve as baselines etc. a “fair comparison” is not what we intend anyway.

L 320 - 322: You state that it should be sufficient to sample epsilon once. But are you also doing that or why is this explicitly mentioned here? Please be more specific.

>> Yes, we are also doing that. Specified in the text.

L. 367 - L. 379: It is not clear from reading the text if these pre-processing procedures are applied on all datasets or not. If not, for which? Furthermore, where do you describe the architectures for the toy model?

>> It is now explicitly mentioned that the pre-processing is applied in all cases. The B-VAE architecture for the toy model is described in L. 353 and following. For the GAN we removed the architecture in a prior revision and now only mention that it’s a Jensen Shannon GAN.

L. 389: What does it mean if a distribution is not correct? What you rather intend to claim is that the generated distribution does not match the reference.

>> Precisely! Fixed.

L. 420 - 421: “Evenly” or “uniformly” distributed? I think you mean uniformly.

>> Fixed.

L. 457: Is it important to state that the values are “comma-separated” in the file?

>> No, removed.

L. 495 - 498: I strongly disagree with the claim that previous works on generative modelling in physics focussed solely on simply comparing histograms. Correlations between reconstructed quantities are also investigated elsewhere. Even in the cited references, Ref. 20 quotes χ^2 for the agreement between observable distributions and Ref. 21 explicitly computes and compares pair-wise correlation coefficients between “real” and “generated” events.

>> corrected

L. 511: Please define what “leading” means.

>> Done

Equation 20: Indexes i, j need to be defined as observable indexes.

>> Done

L. 519: What do you mean by divergences? If you mean alternative definitions, state what those are.

>> Done

L. 523 - 525: Ref. 45 does not suggest the usage of a logarithm. $\chi^2 \rightarrow 0$ in one observable distribution (=supreme accuracy) would lead to $-\infty$. Why do you insist on using it here, especially since the approach without the log does not alter the result?

>> We have removed the log.

L. 555: How are the ideal generator performance indicators obtained?

>> Explained.

L. 574: Is this a general statement or one that applies for the investigated dataset only? If it is the latter, please add "in this study" or write "achieved by using the DijetGAN approach" or similarly.

>> added 'in this study'

L. 581 - 584: This is a repetition from L. 366. Please remove it.

>> modified sentence: only scan on z and B was repetition (architecture/loss), alpha and gamma is new (density buffer).

L 604 - 607: Again, this is a repetition because the pre-processing was described before.

>> Agreed, removed

L. 611: "smearing of events" is a slang which people outside the particle physics analysis community will likely have trouble understanding.

>> We defined the term and continued to use it.

L. 617 - 618: Why are the smeared events close to the true events if all kinematic quantities are smeared by an additional 10% resolution? This is a non-trivial statement.

>> Since we smear the original events by 10% we expect that all 4-vectors are copies (68% CL per quantity) from the original events, but this is of course less trivial for quantities like invariant masses. We have dropped the sentence.

L. 618 - 620: If this procedure does not respect correlations, why does it not perform poorly on Δ_{2d} which itself is essentially a correlation measure?

>> The smearing is done around the original 100.000 events. If the smearing would be done with delta function we would just produce 12 copies from each of the 100.000 events. We expect 2d distributions made from such copies not to be that bad and Δ_{2d} is getting worse for larger smearings.

For very small smearing our other measure Δ_{OF} is expected to get poor (since we produce just copies and expect holes), which is also what we see.

L. 640: As there are no real insights presented and the toy model's training & architecture are not elaborated before, one should consider removing it from this paper.

>> Reasons for keeping the toy model are spelled out in the response to 4). The architecture + training procedure for the B-VAE wrt the toy model is elaborated. For the GAN part we indeed

resorted to only stating that it was a Jensen Shannon GAN, removing the detailed architecture + training in a prior revision.

L. 670: "Plotting" = slang.

>> Changed to 'displaying'

L. 715-716: Claiming that best performances in delta_2D together with poor performances in delta_1D provide small advantages imply a hierarchy among the performance indicators (delta_2D is more important than delta_1D). However, such a hierarchy is not discussed - instead it is said somewhere that they can be weighted arbitrarily.

>> we introduced a hierarchy now and changed the figure of merit.

L. 724: Why do you consider this choice to be the best? You implicitly make use of a weighting of performance indicators hereby which should be quantified somehow.

>> see above.

L. 748: After "Finally" one expects a paragraph/section to end. Yet, the presentation of the results continues afterwards. I would suggest to move this statement on the computing advantage to the end or even in an own subsection.

>> fixed.

L. 769-771: This is a repetition from L. 743-745.

>> fixed.

L. 846-847: What do you consider to be the difference between a good generator and a good density sampler? Why is it not sufficient to only write "good generator of collision data" (or similar)?

>> Changed it to 'good generator of collision data'.

L. 912: "Standard Model cocktail" = slang.

>> fixed.

L. 918 - 934: Please elaborate what the (potential) advantage of using real experimental data for this study is? I.e. what is the added benefit of using CMS open data to your work? A real benefit would be if you compared real CMS data to simulated CMS data to B-VAE generated data.

>> Removed CMS example.

L. 938 - 939: "developed" is a bit too strong as you are mostly deploying standard chi2 measures to compare the histograms. My understanding is that only delta_OF can be considered a non-standard quantity.

>> We now "suggest" instead of develop.

L. 986 - 991: I criticised this statement already before. To me this reads like an inherent contradiction. You essentially explain: "The distribution is not flat due to the low statistics. But if the statistics are increased - it would still not be flat." Furthermore, why would the distribution in phi not be flat for $N_{\text{events}} \rightarrow \infty$? Is this a feature of the underlying CMS detector from Fig. 10?

>> The distribution that is found in the simulated data is not flat for $N=20\,000$. For $N \rightarrow \infty$, we would expect a flat distribution in the simulated data. However producing infinitely many events with our learned model, one would most likely not observe a flat distribution. We would expect that a

model that is trained on a low number of events to learn fluctuations as features. This shows particularly well for small datasets. However, as we removed the CMS example, we also removed this paragraph.

Fig. 7: What does “window size” mean? Do you mean “training iterations”?

>> Clarified: window size of the moving average

REVIEWERS' COMMENTS

Reviewer #1 (Remarks to the Author):

Dear authors,
dear editor,

I acknowledge that my comments to the previous version of this manuscript were properly addressed and I thank the authors for considering the suggested changes to the text.

In my opinion, this manuscript has evolved positively with respect to its first version: The structure has become clearer overall, the performance assessment is more convincing now, and slang was largely eliminated from the text.

At this point, I only have a few remaining remarks which may be incorporated during the eventual typesetting of this paper:

L. 52 - L 55: The definition of z being a random variable is missing.

L. 145: The verb “show” is wrong at this point.

L. 151 and following: The items end with “.” while they end with “,” in the previous paragraph.

L. 169: The MC data are not only training data but also used for evaluation. The word “training” should be dropped.

L. 222: “Dataset” is written “data set” in the rest of the text.

L. 340: unit Gaussian → standard Gaussian

L. 424: “it’s” has a red character in the text

L. 500: “a subset of 10^5 of those samples” reads as if the training sample is less than 10^5 . In order to improve the comprehensibility, one could explicitly mention the splitting ratio of training and evaluation samples.

L. 554: Is this parameterisation, i.e. the function, an empirically found model or is there a theoretical motivation behind it? This should be stated.

L. 633: If I understood the previous passage correctly, an adverb is missing after “generated”.

L. 684: Delta R needs to be defined beforehand.

L. 871: “mapping of latent space” - the statement to what the latent space is mapped to is missing.

L. 911: In the meantime, a work on “Anomaly Detection with Density Estimation” was published elsewhere: <https://dx.doi.org/10.1103/PhysRevD.101.075042> This work should be cited.

L. 955-956: This sentence should be moved to L. 920.

L. 982-988: I had to read this sentence multiple times before fully understanding it (I think). Better formulation:

“All in all, the results of this investigation indicate usefulness of the hereby proposed method not only for particle physics but for all branches of science that involve computationally expensive Monte Carlo simulations, that have the interest to create a generative model from experimental data, or that have the need to sample from high-dimensional and complex distributions.”

L. 1015: The formulation “reasonable request” is strange. What do the authors consider “reasonable”?

Dr. Thorben Quast

Reviewer #1 (Remarks to the Author):

Dear authors,
dear editor,

I acknowledge that my comments to the previous version of this manuscript were properly addressed and I thank the authors for considering the suggested changes to the text.

In my opinion, this manuscript has evolved positively with respect to its first version: The structure has become clearer overall, the performance assessment is more convincing now, and slang was largely eliminated from the text.

>> Dear referee, we thank you for the time, energy and valuable thought that you have put into your reviews and find that the improvement of the manuscript was also a consequence of your efforts.

At this point, I only have a few remaining remarks which may be incorporated during the eventual typesetting of this paper:

L. 52 - L 55: The definition of z being a random variable is missing.

>> definition of z added.

L. 145: The verb "show" is wrong at this point.

>> We agree and removed "show".

L. 151 and following: The items end with "." while they end with "," in the previous paragraph.

>> The items end with "," now.

L. 169: The MC data are not only training data but also used for evaluation. The word "training" should be dropped.

>> The word "training" was dropped.

L. 222: "Dataset" is written "data set" in the rest of the text.

>> Changed to "Data set".

L. 340: unit Gaussian —> standard Gaussian

>> Done.

L. 424: "it's" has a red character in the text

>> Removed and we hope that no other red characters popped up instead.

L. 500: "a subset of 10^5 of those samples" reads as if the training sample is less than 10^5 . In order to improve the comprehensibility, one could explicitly mention the splitting ratio of training and evaluation samples.

>> We improved comprehensibility.

L. 554: Is this parameterisation, i.e. the function, an empirically found model or is there a theoretical motivation behind it? This should be stated.

>> It is now stated.

L. 633: If I understood the previous passage correctly, an adverb is missing after "generated".

>> "sometimes" added.

L. 684: Delta R needs to be defined beforehand.

>> Delta R is now defined.

L. 871: "mapping of latent space" - the statement to what the latent space is mapped to is missing.

>> Reformulated to "2D PCA map of latent space"

L. 911: In the meantime, a work on "Anomaly Detection with Density Estimation" was published elsewhere: <https://dx.doi.org/10.1103/PhysRevD.101.075042> This work should be cited.

>> cited.

L. 955-956: This sentence should be moved to L. 920.

>> Moved to the beginning of the section.

L. 982-988: I had to read this sentence multiple times before fully understanding it (I think). Better formulation:

"All in all, the results of this investigation indicate usefulness of the hereby proposed method not only for particle physics but for all branches of science that involve computationally expensive Monte Carlo simulations, that have the interest to create a generative model from experimental data, or that have the need to sample from high-dimensional and complex distributions."

>> Formulation adopted.

L. 1015: The formulation "reasonable request" is strange. What do the authors consider "reasonable"?

>> "Reasonable" dropped.